# InstaSHAP: Interpretable Additive Models Explain Shapley Values Instantly

**James Enouen**
Department of Computer Science
University of Southern California
Los Angeles, CA
enouen@usc.edu

**Yan Liu**
Department of Computer Science
University of Southern California
Los Angeles, CA
yanliu.cs@usc.edu

## Abstract

In recent years, the Shapley value and SHAP explanations have emerged as one of the most dominant paradigms for providing post-hoc explanations of black-box models. Despite their well-founded theoretical properties, many recent works have focused on the limitations in both their computational efficiency and their representation power. The underlying connection with additive models, however, is left critically under-emphasized in the current literature. In this work, we find that a variational perspective linking GAM models and SHAP explanations is able to provide deep insights into nearly all recent developments. In light of this connection, we borrow in the other direction to develop a new method to train interpretable GAM models which are automatically purified to compute the Shapley value in a single forward pass. Finally, we provide theoretical results showing the limited representation power of GAM models is the same Achilles' heel existing in SHAP and discuss the implications for SHAP's modern usage in CV and NLP.

## 1 Introduction

Since their introduction into machine learning, Shapley values have had a meteoric rise within the space of model explanation. The principled axioms of Shapley (Shapley, 1953) and the easy-to-use framework of SHAP (Lundberg & Lee, 2017) have led to their widespread adoption when compared with alternatives in gradient-based and black-box explanation methods. The developments which then followed quickly pushed beyond tabular datasets into higher dimensional data like computer vision and natural language, with abundant research investigating how to improve the speed and efficiency of Shapley values across these various high-dimensional domains (Covert et al., 2021; Mosca et al., 2022; Jethani et al., 2022; Covert et al., 2023; Enouen et al., 2024). In recent years, however, some lines of work have identified specific application scenarios where the Shapley value is provably guaranteed to fail, perhaps begging the question of whether such works improving on the Shapley value are instead done in vain (Bilodeau et al., 2022; Huang & Marques-Silva, 2023). Unfortunately, many of these critiques have been made piecemeal without an overall sense of their underlying causes. In contrast, this work takes the perspective that SHAP's issues of explanation speed and explanation power can all be viewed under the same lens through the underlying connection with additive models and feature interactions.

In this work, we find that all of the most recent development in the SHAP value, like the practical improvements of FastSHAP (Jethani et al., 2022) and the theoretical advancements of FaithSHAP (Tsai et al., 2023), can be unified and more easily understood by using a functional perspective using additive models. In particular, the amortization scheme introduced by FastSHAP (Jethani et al., 2022) builds on the least squares formulation of the Shapley value (Charnes et al., 1988; Lundberg & Lee, 2017) by training a global approximator for the Shapley value, which can each be viewed as fitting a pointwise and a global additive model, respectively. Tsai et al. (2023) instead extends this least squares characterization to the bivariate and higher-order interactions to yield a richer understanding of the model to explain, drawing the same parallels with higher-order additive models.

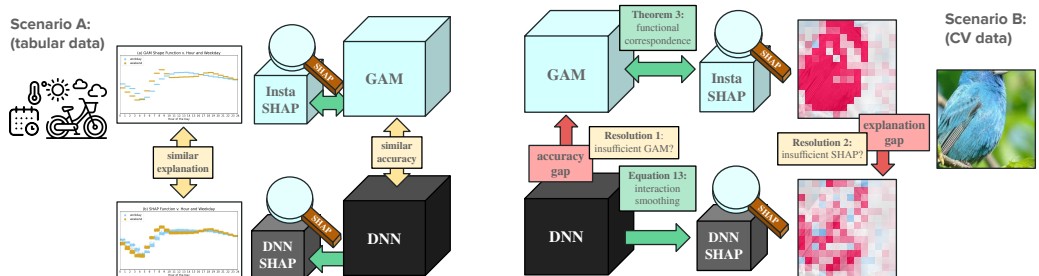

Figure 1: The fundamental correspondence between SHAP and GAM is used practically to distinguish two unique scenarios. In scenario A, such as simpler tabular data, GAM models can achieve SOTA performance and their SHAP explanations align with SHAP explanations of blackbox models. In modern ML applications like computer vision, we have scenario B, where there is a gap between GAM and DNN performance in practice. This means that either: (#1) we cannot train GAMs as well as other deep neural networks; or (#2) there is an overcredulous trust of SHAP in these domains.

All of these extensions and more can be though of as special cases of training additive models when considered from the functional ANOVA perspective (Sobol, 2001), allowing not only for their simple combination but a greater understanding of the underlying mechanisms overall. We further find that there is a deep variational connection which underscores all such recent developments between the fundamentally interrelated SHAP explanation and GAM model, paralleling the algebraic case studied in Bordt & von Luxburg (2023) and building on various works in the GAM literature studying purification of additive models (Hooker, 2007; Hart & Gremaud, 2018; Lengerich et al., 2020; Sun et al., 2022).

Overall, we set out to emphasize these fundamental connections between SHAP and GAM (as well as Faith-SHAP-k and GAM-k) across all possible correlated input distributions, where previous work only addressed the case of independent input variables (Bordt & von Luxburg, 2023). We use these theoretical advances in the variational equation to provide a simple check of whether SHAP is trustworthy by requiring a GAM model can be trained to the same accuracy (as depicted in Figure 1.) This development allows for deeper insights and practical checks into SHAP's application in important ML domains like CV and NLP where input features are heavily correlated. Further, by developing a new technique to automatically purify additive models to return the corresponding SHAP values, we simultaneously solve a longstanding problem from the GAM literature as well as demonstrate practical advantages of our method InstaSHAP over the existing FastSHAP. We envision our major contributions as follows:

- Establishment of the variational formulation of Shapley additive models alongside the existing variational formulations of GAM and functional ANOVA, solving the case of dependent variables which was left open in Bordt & von Luxburg (2023).

- Introduction of a practical training method for GAM models via an alternative loss function with output masking, automatically solving the problem of GAM purification and allowing for 'instant' access to the Shapley values.

- Theoretical insights into the real-world application of SHAP and many comparative experiments across synthetic and real-world domains of interest, helping to lay bare the question of whether Shapley values are trustworthy for ML in practical environments.

## 2 BACKGROUND

Let $F : \mathcal{X} \to \mathcal{Y}$ be a function representing a machine learning model which maps from input space $\mathcal{X} \subseteq \mathbb{R}^d$ to output space $\mathcal{Y} \subseteq \mathbb{R}^c$, where there are $d$ input features and $c$ output features. We will use $[d] := \{1, \ldots, d\}$ to represent the set of input features and $S \subseteq [d]$ to represent a subset of the input features. We also write the set of all such subsets, the powerset, as $\mathcal{P}([d]) \cong \{0, 1\}^d$.

## 2.1 EXPLAINING BY REMOVING

A very important aspect of removal-based explanations like LIME or SHAP (Ribeiro et al., 2016; Lundberg & Lee, 2017) is the method of feature removal (Sundararajan & Najmi, 2020; Covert et al., 2021). We review the three most popular removal approaches: replacing the feature with a reference value (baseline), integrating over the marginal distribution of the feature (marginal), or integrating over the conditional distribution of the feature (conditional). When applied to an explanation method like the Shapley value, these result in the corresponding: baseline Shapley, marginal Shapley, or conditional Shapley (Sundararajan & Najmi, 2020; Janzing et al., 2020; Frye et al., 2021).

First, consider an input example one would like to explain $x \in \mathcal{X}$ and a subset of the features which one would like to keep $S \subseteq [d]$ as part of the model. We may then compare against a baseline value $\bar{x} \in \mathbb{R}^d$ and define the baseline value as $\mathcal{B}_{\bar{x}}$ as below. We may also choose a distribution of baselines $p(x)$ over the input space $\mathcal{X}$, which allows us to define both the marginal projection and the conditional projection, $\mathcal{N}_p$ and $\mathcal{M}_p$.

$$[\mathcal{B}_{\bar{x}} \circ F](x, S) := F(x_S, \bar{x}_{-S}) \tag{1}$$

$$[\mathcal{N}_p \circ F](x, S) := \mathbb{E}_{\bar{X}_{-S} \sim p(X_{-S})} \left[ F(x_S, \bar{X}_{-S}) \right] \tag{2}$$

$$[\mathcal{M}_p \circ F](x, S) := \mathbb{E}_{\bar{X}_{-S} \sim p(X_{-S}|X_S=x_S)} \left[ F(x_S, \bar{X}_{-S}) \right] \tag{3}$$

We write these three operators as functionals mapping $F$ to a new function $f$ to support our analysis from a functional perspective. Historically, the baseline value and marginal value have been the easiest to use in practice because we may directly explain our blackbox $F$ without significant modifications. However, since the highlighting of the 'off-the-manifold' problem by Frye et al. (2021), it has been shown that baseline methods $\mathcal{B}_{\bar{x}}$ and marginal methods $\mathcal{N}_p$ significantly overemphasize the algebraic structure of the model instead of the statistical structure. If one is exclusively interested in the algebraic dependencies of their ML model, the correspondence highlighted herein has already been established in Bordt & von Luxburg (2023). Otherwise we hereafter restrict our attention to the conditional expectation using $\mathcal{M}_p$ and provide details on further considerations in Appendix A.1.

We define a feature attribution method $\Phi$ as taking $F(x)$ and returning a local explanation function $[\Phi_i \circ F](x)$ for each feature $i \in [d]$ on each local input $x \in \mathcal{X}$. Similarly, we define a blackbox feature attribution method as instead taking a masked function $f(x, S)$ and returning a local explanation function for each feature, $[\phi_i \circ f](x)$.

We now provide one of the typical definitions of the Shapley value as follows; however, we recommend the unfamiliar reader instead waits until the more intuitive Equation 13.

$$[\phi_i^{\text{SHAP}} \circ f](x) = \sum_{S \subseteq [d]} p^{\text{SHAP-unif}}(S) \cdot \left[ f(x, S + i) - f(x, S - i) \right] \tag{4}$$

$$p^{\text{SHAP-unif}}(S) \propto \binom{d}{s}^{-1} \frac{1}{d+1} \tag{5}$$

Here, the Shapley value is defined as the addition and removal of a single feature $i \in [d]$ across many contexts $S \subseteq [d]$ according to the distribution $p^{\text{SHAP-unif}}(S)$ where the shorthand $s = |S|$ is used. Following the discussion in the previous section, we will in this work always consider the conditional Shapley $\Phi^{\text{cond-SHAP}} := \phi^{\text{SHAP}} \circ \mathcal{M}_p$. Alternative black-box explanations to the Shapley value are discussed further in Appendix A.2.

## 2.2 INTERPRETING BY ADDING

We now introduce the interpretable generalized additive model (GAM) of Hastie & Tibshirani (1990). This model is seen as interpretable because each of the input features have a simple 1D relationship with their effect on the output. In this work, we also include the 'zero dimensional' normalizing constant $f_\emptyset$ and often use the term GAM1 to emphasize a GAM that only has 1D functions.

$$
\begin{aligned}
F^{\leq 1}(x_1, \ldots, x_d) \;&=\; f_\emptyset \;+\; \underbrace{f_1(x_1) + \cdots + f_d(x_d)} \\
&=\; f_\emptyset \;+\; \underbrace{\sum_{i \in [d]} f_i(x_i)}
\end{aligned}
\tag{6}
$$

This can further be generalized to a GAM2 model (Wahba et al., 1994; Lou et al., 2012; 2013; Chang et al., 2022) which is still seen as an interpretable model because its 2D functions can still be visualized using a heatmap plot.

$$
\begin{aligned}
F^{\leq 2}(x_1, \ldots, x_d) \;&=\; f_\emptyset \;+\; \underbrace{f_1(x_1) + \cdots + f_d(x_d)} \;+\; \underbrace{f_{1,2}(x_1, x_2) + \cdots + f_{d-1,d}(x_{d-1}, x_d)} \\
&=\; f_\emptyset \;+\; \underbrace{\sum_{i \in [d]} f_i(x_i)} \;+\; \underbrace{\sum_{\{i,j\} \subseteq [d]} f_{i,j}(x_i, x_j)}
\end{aligned}
\tag{7}
$$

Recent research has additionally focused on addressing the practical considerations associated with training increasingly high-order GAMs (Yang et al., 2020; Dubey et al., 2022; Enouen & Liu, 2022). For some order $k \geq 3$, we may define the higher-order GAM-k as:

$$
F^{\leq k}(x_1, \ldots, x_d) \;=\; f_\emptyset \;+\; \sum_{i \in [d]} f_i(x_i) + \; \ldots \; + \sum_{S \subseteq [d], |S| = k} f_S(x_S) \;=\; \sum_{S \in \mathcal{I}_{\leq k}} f_S(x_S)
\tag{8}
$$

where we write $\mathcal{I}_{\leq k} := \{S \subseteq [d] : |S| \leq k\}$.

This might immediately raise the question of when to stop adding higher-order functions to our GAM model. Multiple practical works have shown that for tabular data, $k$ does not have to be too large: GAM-1 and GAM-2 are often sufficient to fit the complexities of the data and achieve state-of-the-art performance across many datasets (Chang et al., 2022; Enouen & Liu, 2022). The same question for CV or NLP, however, has faced little exploration in previous works. In order to answer this question precisely, however, we instead turn to the functional ANOVA decomposition coming from the field of sensitivity analysis.

## 2.3 FUNCTIONAL ANOVA

In the literature on sensitivity analysis, we may take any function and completely decompose it via its ***functional ANOVA decomposition*** (Sobol, 2001; Hooker, 2004):

$$
F(x_1, \ldots, x_d) \;=\; \sum_{S \subseteq [d]} \tilde{f}_S(x_S)
\tag{9}
$$

Although there are many possible choices of $\tilde{f}_S$ which could obey this equation, we may define a unique decomposition via the conditional projection from Section 2.1:

$$
\tilde{f}_S(x_S) := \sum_{T \subseteq S} (-1)^{|S| - |T|} f(x, T) = \sum_{T \subseteq S} (-1)^{|S| - |T|} [\mathcal{M}_p \circ F](x, T)
\tag{10}
$$

Hereafter, we often follow the sensitivity analysis notation of writing $\tilde{f}_S(x_S)$ rather than $\tilde{f}(x, S)$.

This specific choice using conditional projection is often called the 'Sobol-Hoeffding' decomposition. In the case of independent input variables, Sobol (2001) provides us a complete understanding of what happens to the variational solution of training any additive model. In particular, the variance or the mean squared error is able to decompose completely via the ***decomposition of variance*** formula:

$$
\mathbb{V} := \mathbb{V}\mathrm{ar}_X[F(X)] = \sum_{S \subseteq [d]} \mathbb{V}\mathrm{ar}_{X_S}[\tilde{f}_S(X_S)] = \sum_{S \subseteq [d]} \mathbb{V}_S
\tag{11}
$$

where the orthogonal contributions, $\mathbb{V}_S := \mathbb{V}\mathrm{ar}_{X_S}[\tilde{f}_S(X_S)]$, are called the Sobol indices and measure the variance which can be uniquely ascribed to each feature interaction $S$.

Unfortunately, this variational formulation for additive models breaks down for the case of correlated input variables. The best existing alternative in the literature is the Sobol covariances (Rabitz, 2010; Hart & Gremaud, 2018) which is instead defined as $\mathbb{C}_S := \mathbb{C}\mathrm{ov}_X[F(X), \tilde{f}_S(X_S)]$. However, these

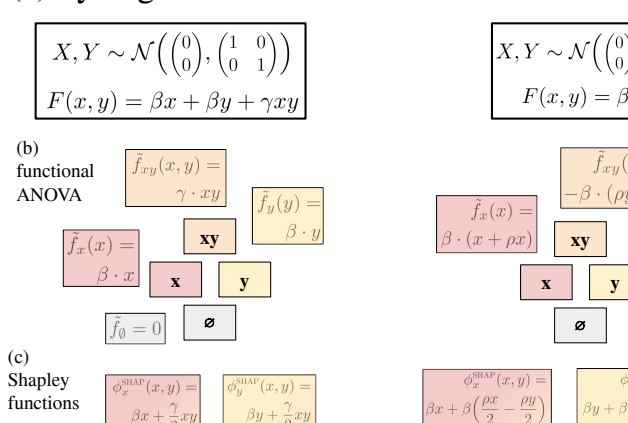

Figure 2: Simple examples (Gaussian input variables and multilinear response variables) which demonstrate each of the two major types of feature interactions: synergistic interactions and redundant interactions. Their full functional ANOVA and exact Shapley functions are additionally calculated and shown. Colored by relevant variable. Note we use $x$ and $y$ instead of $x_1$ and $x_2$.

covariances may result in values which are both positive and negative, conflating the synergistic effects between a set of features $S$ and the redundant effects of shared information amongst the same set of features $S$.

We will still say that a statistical ***feature interaction***, $S \subseteq [d]$, is said to exist whenever

$$\mathbb{V}_S := \mathbb{Var}_{X_S}[\tilde{f}_S(X_S)] > 0 \ , \tag{12}$$

however, for the case of correlated input features, we importantly need to distinguish between the two major types of feature interactions:

1. feature interaction *synergy*, where $\mathbb{V}_S > 0$ and $\mathbb{C}_S > 0$.
2. feature interaction *redundancy*, where $\mathbb{V}_S > 0$ but $\mathbb{C}_S < 0$.

Using this functional ANOVA perspective, we can now write the Shapley value as a complete function for each variable $i \in [d]$ using the well known alternative via the synergy or unanimity functions (Shapley, 1953):

$$[\phi_i^{\text{Sh}} \circ f](x) = \sum_{S \supseteq \{i\}} \frac{\tilde{f}_S(x_S)}{|S|} \tag{13}$$

Intuitively, the value which is ascribed to each feature interaction $S$ is uniformly divided amongst each of its constituents $i \in S$. In Figure 2, we can see the easily computed Shapley functions coming from computing the functional ANOVA decomposition, dividing the interaction effects in both the synergistic and the redundant setting. In real-world datasets and in the presence of higher-order interactions, it easy to imagine how quickly such effects will compound and conflate one another.

## 3 REPRESENTATION POWER OF ADDITIVE MODELS

Before proceeding further with the variational GAM methods we introduce, we find it is important to characterize the behavior of SHAP in terms of the functional ANOVA decomposition. In particular, we will do this in the form of an "impossibility theorem" to help cement the correspondence which exists between GAM and SHAP. However, unlike previous works focusing on the flaws of SHAP, we not only exactly characterize all negative results showing when hypothesis tests are impossible, but consequently characterize all positive results showing exactly when hypothesis tests are possible.

## 3.1 SHAP FUNCTION SPACE

**Theorem 1. (SHAP≅ANOVA-1)** SHAP will succeed on any hypothesis for some hypothesis space $\mathcal{H}$ if and only if $\mathcal{H}$ is completely free of feature interactions ($\mathcal{H} \subseteq \mathcal{H}_{\text{ANOVA}}^{\leq 1}$).

We leave the details of the proof until Appendix C; however, for one direction it is relatively straightforward to see from from Equation 13 that SHAP can succeed if all interaction terms are zero. Conversely, if some true model $F \in \mathcal{H}$ is not representable by an ANOVA-1 model, then $F \notin \mathcal{H}_{\text{ANOVA}}^{\leq 1}$ and hence SHAP is instead obscuring the feature interactions. Importantly, we emphasize that this means SHAP cannot distinguish synergistic feature interactions nor can it distinguish redundant feature interactions. We can additionally show the exact same type of relationship is true for Faith-SHAP-k.

**Theorem 2. (Faith-SHAP-k≅ANOVA-k)** For any $k \in [d]$, Faith-SHAP-k will succeed on any hypothesis test in some hypothesis space $\mathcal{H}$ if and only if $\mathcal{H}$ is free of higher-order features interactions of size $(k + 1)$ or greater ($\mathcal{H} \subseteq \mathcal{H}_{\text{ANOVA}}^{\leq k}$).

This similarly implies that even indices measuring feature interactions will still be forced to blur out the higher-order interactions and hence remain limited in their representational capacity. Once again, we emphasize that the reliance of these approaches on the functional ANOVA decomposition means it is not possible for them to distinguish between synergistic interactions and redundant interactions.

## 3.2 GAM FUNCTION SPACE

Let us now contrast these two results with the representation power of GAM models.

**Theorem 3. (ANOVA-1⊊GAM-1)** The functional space of ANOVA-1 representable functions is a strict subset of the functional space of GAM-1 representable functions ($\mathcal{H}_{\text{ANOVA}}^{\leq 1} \subsetneq \mathcal{H}_{\text{GAM}}^{\leq 1}$).

Any function which is representable by a univariate ANOVA decomposition is automatically representable by a GAM model by the subset compliance of the ANOVA decomposition. This inclusion is strict in the other direction as soon as there is a feature correlation in the input data.

**Theorem 4. (ANOVA-k⊊GAM-k)** The functional space of ANOVA-k representable functions is a strict subset of the functional space of GAM-k representable functions ($\mathcal{H}_{\text{ANOVA}}^{\leq k} \subsetneq \mathcal{H}_{\text{GAM}}^{\leq k}$).

Once again, the inclusion of k-dimensional ANOVA functions are automatically representable by an arbitrary GAM-k model by the definition of the decomposition. The inclusion is again strict as soon as there is a correlation between features in the input data. We save proofs and further discussions for Appendix C.

## 3.3 PRACTICAL INSIGHTS

In conclusion, our results show that a practitioner may evaluate the trustworthiness of SHAP on a given dataset by simply training a GAM model on the same dataset. If a GAM can easily match the same accuracy as a blackbox model or easily distill the same predictions as a blackbox model, then this is a dataset for which SHAP explanations can generally be trusted. On the other hand, if a GAM cannot match the same accuracy as the blackbox model, this means that the practitioner needs to be wary about trusting SHAP values on this dataset. In this second scenario, there are two possible resolutions. For the GAM researcher, resolution 1 of Figure 1 is to train a better GAM through the use of more efficient training procedures or through an increase in capacity with additional feature interaction terms. For the SHAP practitioner, resolution 2 of Figure 1 is to admit that SHAP is likely not a sufficient explanation for this model or dataset.

In many real-world scenarios, it is possible that neither of these extremes is completely true, with the lower bound GAM and upper bound SHAP meeting somewhere in the middle. Nevertheless, in the current literature, this gap is extremely large for practical AI tasks including CV and NLP. In our experiments, we highlight this large gap on a high-dimensional CV task of bird classification. Ultimately, the key contribution of this theory is that it provides a practical test for researchers to understand task-by-task what are the advantages as well as the limitations of SHAP and GAM approaches.

## 4   SHAPLEY VIA LEAST SQUARES

**Kernel SHAP**   The Shapley value was first given an optimization-based definition or 'variational' characterization in Charnes et al. (1988) as the solution to:

$$\underset{\phi \in \mathbb{R}^d}{\arg\min} \left\{ \mathbb{E}_{S \sim p^{\text{SHAP}}(S)} \left[ \left| f(S) - \sum_{i=1}^d \mathbb{1}(i \in S) \cdot \phi_i \right|^2 \right] \right\} \tag{14}$$

where the distribution is over the SHAP kernel

$$p^{\text{SHAP}}(S) \propto \binom{d}{s}^{-1} \frac{1}{s(d-s)} \tag{15}$$

where once again $s = |S|$ (contrast this distribution with Equation 5). This formulation was originally utilized by KernelSHAP (Lundberg & Lee, 2017) to fit a local linear model according to the SHAP kernel distribution in order to attain sufficient speed to see use in ML applications.

**Fast SHAP**   This was importantly used more recently by FastSHAP (Jethani et al., 2022) in order to create a functional amortization scheme which fits to the same SHAP kernel. They then write the Shapley function as the solution to the following equation:[1]

$$\underset{\phi:\mathcal{X} \to \mathbb{R}^d}{\arg\min} \left\{ \mathbb{E}_{x \sim p(x)} \left[ \mathbb{E}_{S \sim p^{\text{SHAP}}(S)} \left[ \left| f(x; S) - \sum_{i=1}^d \mathbb{1}(i \in S) \cdot \phi_i(x) \right|^2 \right] \right] \right\} \tag{16}$$

where they then fit the functions $\phi_i : \mathcal{X} \to \mathbb{R}$ over the entire input space to automatically generate the Shapley values at test time. This dramatically improves the test-time speed with which SHAP explanations can be generated, overcoming what is often the most major practical limitation to deployment.

The summing over multiple functions to create the predicted output should remind the reader of the structure of GAM-1 additive models. As we saw in our impossibility theorems and as we will later show with InstaSHAP, this functional perspective indeed opens up the possibility to connect with training additive models.

**Faith SHAP**   As we highlight in Theorem 1, it is well known that feature interactions are necessary for explaining more complex functions. Accordingly, many works have tried to extend the Shapley value to be able to handle the effects of feature interactions (Grabisch & Roubens, 1999; Sundararajan et al., 2020; Bordt & von Luxburg, 2023; Fumagalli et al., 2023). Recently, there have been theoretical advancements which extend the Shapley value directly using the variational formulation in Tsai et al. (2023). They write their higher-order solution, called Faith SHAP, as:

$$\underset{\phi \in \mathbb{R}^{\mathcal{I}_{\leq k}}}{\arg\min} \left\{ \mathbb{E}_{S \sim p^{\text{SHAP}}(S)} \left[ \left| f(S) - \sum_{T \subseteq [d], |T| \leq k} \mathbb{1}(T \subseteq S) \cdot \phi_T \right|^2 \right] \right\} \tag{17}$$

Once again, we can see parallels between the GAM-k model from Equation 8. The case of $k = 1$ indeed reduces to the original least squares solution in Equation 14.

## 5   INSTANT SHAP

A simple combination of these two ideas (functional amortization and feature interaction) would lead to an explainer which automatically recovers the two-dimensional or higher-dimensional Faith-SHAP explanations of the target function, while maintaining the practical speedups of FastSHAP:

$$\underset{\phi:\mathcal{X} \to \mathbb{R}^{\mathcal{I}_{\leq k}}}{\arg\min} \left\{ \mathbb{E}_{x \sim p(x)} \left[ \mathbb{E}_{S \sim p^{\text{SHAP}}(S)} \left[ \left\| f(x; S) - \sum_{T \subseteq [d], |T| \leq k} \mathbb{1}(T \subseteq S) \cdot \phi_T(x) \right\|^2 \right] \right] \right\} \tag{18}$$

---

[1]To keep the notation cleaner and more similar with other existing works, we assume throughout this section that $f_\emptyset = 0$, which is equivalent to centering or normalizing the outputs.

This functional amortization automatically recovers the Faith-SHAP-k values as defined in Tsai et al. (2023), but maintains the practical advantages and speedups of FastSHAP from Jethani et al. (2022).

Instead, however, we propose to adapt the typical variational equations used to fit GAM models to fall under this same variational Shapley framework. We first make explicit the variational equation used to train GAM models as:

$$\underset{\phi:\mathcal{X}\rightarrow\mathbb{R}^{\mathcal{I}_{\leq k}}}{\arg\min}\left\{\mathbb{E}_{x\sim p(x)}\left[\mathbb{E}_{S\sim p^{\text{GAM}}(S)}\left[\left\|f(x;S)-\sum_{T\subseteq[d],|T|\leq k}\phi_T(x_T)\right\|^2\right]\right]\right\} \quad (19)$$

Accordingly, we define the Insta-SHAP-GAM as an additive model which is trained as:

$$\underset{\phi:\mathcal{X}\rightarrow\mathbb{R}^{\mathcal{I}_{\leq k}}}{\arg\min}\left\{\mathbb{E}_{x\sim p(x)}\left[\mathbb{E}_{S\sim p^{\text{SHAP}}(S)}\left[\left\|f(x;S)-\sum_{T\subseteq[d],|T|\leq k}\mathbb{1}(T\subseteq S)\cdot\phi_T(x_T)\right\|^2\right]\right]\right\} \quad (20)$$

where first we introduce the masked training objective where $S$ are drawn from the Shapley kernel and second we add the masking on each additive component of the GAM to only be included if all features of that component are unmasked in $S$.

This new formulation is able to bring novel insights to both the literature on SHAP and the literature on GAM. For SHAP, we incorporate the low-dimensional GAM bias which is able to more accurately model SHAP values when compared with FastSHAP. Additionally, by the explicit modeling of interactions, we are able to improve on the practical feasability of methods like FaithSHAP, which have yet to develop a method for practical deployment. For the GAM literature, we make progress towards the longstanding goal of purification of the shape functions of additive models. In the appendix, we further detail how this extends on the existing GAM literature and its various applications towards the selection of feature interactions under correlated inputs.

# 6 TABULAR EXPERIMENTS

## 6.1 SYNTHETIC EXPERIMENTS

We construct a simple dataset to test the varying effects of synergistic interactions and redundant interactions in a similar spirit to Figure 2. We provide additional details in the Appendix, but we use a simple ten-dimensional feature space with an algebraic GAM rank of $k*$ and a correlation of $\rho$. In Figure 3, we see the approximation results consistently showing InstaSHAP has a better inductive bias than FastSHAP for learning the Shapley values.

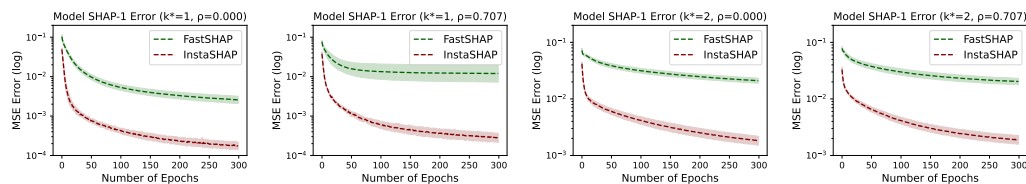

Figure 3: MSE error of approximations of the model's SHAP values. Since both FastSHAP and InstaSHAP are functional approximations, we report the MSE errors across the epochs of training.

## 6.2 SYNERGY IN BIKE SHARING

In the bike sharing dataset, we find strong evidence of a synergistic interaction effect. This dataset predicts the expected bike demand each hour given some relevant features like the day of the week, time of day, and current weather. There is a total of thirteen different input features predicting a single continuous output variable.

In the case of a multi-layer perceptron trained to predict the bike demand, the normalized mean-squared error ($R^2$) results is $6.59\%$. Although a GAM-1 can only achieve an $R^2$ of $17.4\%$, a low-dimensional GAM can achieve an $R^2$ of $6.23\%$. It is well-known that on this dataset there is a strong interaction between the hour variable and workday variable (since people's schedules change

on the weekend vs. a workday.) In Figure 4a, we can see how this feature interaction is adequately captured. In Figure 4b, we can see how if the SHAP value also makes this trend identifiable, but does not completely separate this bivariate interaction from other feature effects in the dataset. Overall, Figure 4 supports the hypothesis that training interpretable models is a better path than explaining blackbox models, especially when the same accuracies can be achieved.

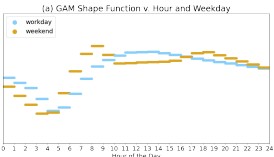 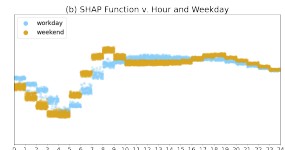 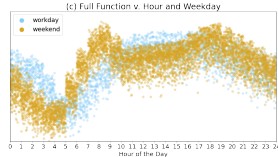

Figure 4: The Spectrum of Interpretability to Uninterpretability. We display the key {hour, workday} interaction for the interpretable GAM, explainable SHAP, and uninterpretable blackbox.

### 6.3 REDUNDANCY IN TREE COVER

In the treecover dataset, we find strong evidence of a redundant interaction effect. This dataset consists of predicting one of the seven types of tree species which are covering a given plot of land based on eleven input features describing the area. Simple investigation can determine the most important features for determining the species of tree are the altitude of the land and soil category of the land. Accordingly, we provide their partial dependence plots in Figure 5a and 5b.

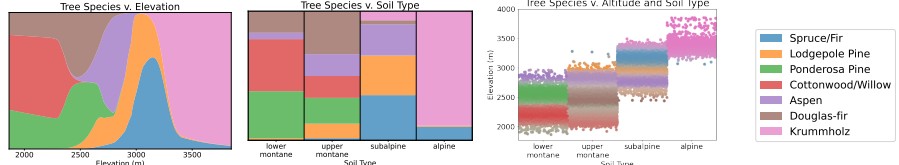

Figure 5: 1D and 2D Dependence of Tree Species on Altitude and Soil

However, a cursory look at the 1D dependence such as these plots or SHAP ignores the fact that both the elevation and soil type are additionally correlated with one another. Indeed, some montane-type soils can only be found in lower altitudes and, equally, alpine-type soils can only be found at higher altitudes. Looking at the 2D heatmap in Figure 5c, we can see that soil and altitude are correlated with one another and somewhat redundantly predict the joint trend in the species of tree.

Training an MLP on this dataset is able to achieve validation accuracy of $80.4\%$ whereas a GAM-1 can only achieve $72.4\%$ accuracy. Alternatively, a low-dimensional GAM is instead able to achieve $82.2\%$ accuracy. This once again demonstrates that although the 1D SHAP is unable to accurately represent this tabular dataset, a simple low-dimensional GAM is able to as well.

Beyond the datasets we study here, there are many existing works showing that both (a) feature interactions are necessary for real-world datasets, and (b) relatively low-dimensional GAM models can often achieve SOTA performance on tabular datasets (Chang et al., 2022; Enouen & Liu, 2022). Accordingly, after the clear demonstration of both categories (synergy and redundancy) of feature interactions in practice, we move on to exploring the same phenomenon in higher-dimensional data.

## 7 HIGHER DIMENSIONAL EXPERIMENTS

We additionally use our methods to explore a bird classification task on natural images. We run experiments on the CUB dataset using a resnet CNN architecture, evaluating not only at the original, fine-grained 200 classes corresponding to each bird's species, but also with 37 coarse-grained classes corresponding to each bird's family. For our GAM models we adapt the resnet architecture to only include the influence from a $1 \times 1$, $2 \times 2$, or $3 \times 3$ set of patches, leading to a GAM-1, GAM-4, and GAM-9 model. We fine-tune all models with procedures similar to Covert et al. (2023), see appendix for details.

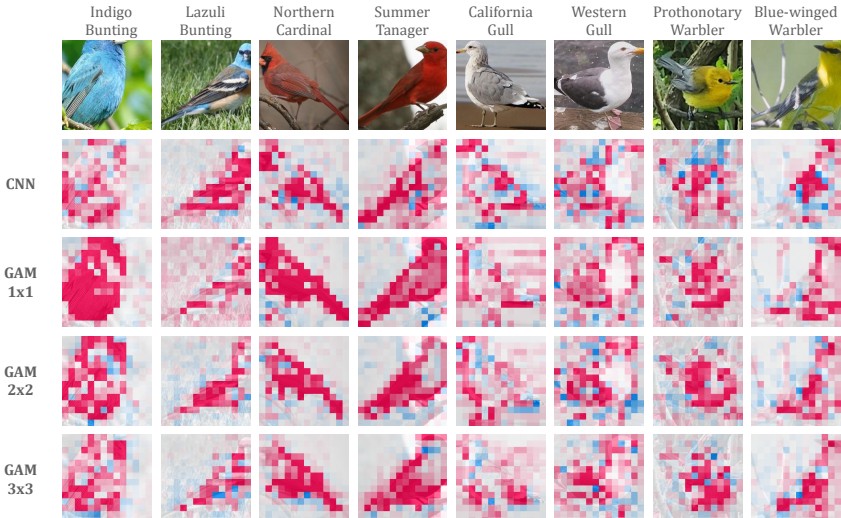

Figure 6: Display of SHAP explanations for multiple images inside of the CUB birds dataset. Explanations are provided for the CNN model as well as multiple GAM models. We can see that beyond the gap in accuracy between the CNN and GAM models, there are also discrepancies in the reasoning processes as explained by SHAP. This can be taken as strong evidence of oversmoothing of the interaction effects which are used by the original CNN model.

Fine-tuning a resnet-50 model on the dataset, we are able to achieve a fine-grained accuracy of $65.0\%$ and a coarse-grained accuracy of $81.8\%$. In contrast, the GAM-1 model is only able to achieve a $33.2\%$ fine and $53.7\%$ coarse accuracies. This can be taken as strong evidence that SHAP is oversimplifying the behavior of the CNN on this dataset. In Figure 6, we can see that this also manifests as sizable visual differences in the explanations between the two models. In some sense, at least 20 percentage points of accuracy are being completely thrown away when using the simplification of SHAP. So although a useful first approximation of the understanding, SHAP is once again not sufficient for giving a complete understanding of the behavior of feature interactions.

Extension to the GAM-2x2 and GAM-3x3 models is able to give improvements to $45.8\%$ and $46.8\%$ fine as well as $66.3\%$ and $66.8\%$ coarse accuracies. Even with some feature interactions, a convolutional GAM model is unable to achieve the same accuracies as a resnet. This points to the fact that either long-range or higher-order interactions are necessary to completely match the performance of the resnet. It is also possible that an alternate GAM architecture would be able to further improve upon these accuracy results, helping close the gap between CNN and GAM. Overall, we take this as significant evidence towards the oversimplification of SHAP explanations on high-dimensional data, domains where the need for explainability also remains the highest.

## 8 CONCLUSION

We find that the study of SHAP and GAM from a joint functional perspective allows for a plethora of insights in both domains which were not previously possible. We establish the theoretical correspondence between the two across all possible correlated input features and discuss the implications in terms of functional representation power. In practical ML datasets where input correlations are abundant, we provide a simple but theoretically grounded method of detecting whether SHAP is providing adeqeuate explanations by means of training a GAM model. We extend on the GAM literature by means of rigorously studying the effect of training on a correlated input distribution, as well as introducing a novel masking technique which allows for the recovery of purified GAM models. In multiple real-world datasets, we find that the existence of feature interactions as synergies and as redundancies is ubiquitous in practical settings. We finally discuss the implications of this fact in the context of interpreting SHAP in high-dimensional data like natural images. Although SHAP is a very useful approximation of the first-order effects, a more careful treatment of feature interactions will be required for a complete understanding of blackbox models.

## 9 BROADER IMPACT

This work focuses on enhancing the interpretability of deep learning models. Although, broadly, the work of interpretability can help inform all related stakeholders to the reasonings behind decisions made by AI systems to the benefit of everyone involved, ultimately, all interpretations and decisions are made by humans and can hence be used for unfavorable outcomes both intentionally and unintentionally. Moreover, interpretability is only one piece of the larger puzzle which is transparency and trustworthiness in AI systems.

## 10 ACKNOWLEDGEMENTS

This work was supported in part by the Department of Defense under Cooperative Agreement Number W911NF-24-2-0133. The views and conclusions contained in this document are those of the authors and should not be interpreted as representing the official policies, either expressed or implied, of the Army Research Office or the U.S. Government. The U.S. Government is authorized to reproduce and distribute reprints for Government purposes notwithstanding any copyright notation herein.

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

# A  POST-HOC EXPLAINABILITY

## A.1  FURTHER DISCUSSION OF EXPLANATION BASELINE METHODS

**Conditional/ Marginal/ Baseline**   We first reiterate the three main removal baselines which see widespread usage across all domains of machine learning explainability. Those are the three methods introduced in the main text (baseline value, marginal value, and conditional value.) It should be noted for instance that in the original SHAP paper (Lundberg & Lee (2017), Equations 9-12), each of the former two were considered as a simplification or approximation to the conditional value. The first assumption of feature independence implies the equivalence of the conditional value and the marginal value. The second assumption of model linearity implies the equivalence of the marginal value and the baseline value. Accordingly, it is perhaps better to think of these two alternatives as practical simplifications whereas the conditional value is the value of theoretical interest. Especially after the highlighting of the off-the-manifold problem (Frye et al., 2021), these two approaches have been under higher scrutiny in their application to typical ML pipelines where input data often have heavy correlations existing outside of the control of the ML practitioner.

**Integrated Gradients**   Another common removal approach is Integrated Gradients which is equivalent to Aumann-Shapley value (Auman & Shapley, 1974; Sundararajan et al., 2017; Sundararajan & Najmi, 2020). In this version, a line integral is taken from a baseline point $\bar{x}$ to the target point $x$, rather than the original baseline method which simply takes the difference between the two. Although it is a smoother approximation which has had empirical success, its interaction extensions (Janizek et al., 2021) cannot succeed on non-smooth functions like piecewise linear ReLU networks and it is nonetheless susceptible to the off-the-manifold problem. Although incorporating a more general definition of line integrals could be of interest to solving the off-the-manifold and simultaneously integrating into the discrete masking framework we utilize, we envision this as out of scope for our focus on the Shapley value.

**Stone-Hooker Decomposition**   Of potentially the greatest interest besides the conditional case which we directly study is the alternative functional ANOVA decomposition proposed by Hooker (2007) and further investigated in (Hart & Gremaud, 2018; Lengerich et al., 2020; Sun et al., 2022).

$$F(x_1, \ldots, x_d) \;\; = \;\; \sum_{S \subseteq [d]} \tilde{h}_S(x_S) \tag{21}$$

where the functions are required to obey a set of 'hierarchical orthogonality conditions'

$$\mathcal{M}_{p,\emptyset} \circ (g_T \cdot \tilde{h}_S) = 0 \qquad \forall g_T, \;\; \forall T \subsetneq S \tag{22}$$

which is equivalent to the 'integral conditions'

$$\mathcal{M}_{p,(S-i)} \circ (\tilde{h}_S) \equiv 0 \qquad \forall i \in S. \tag{23}$$

Despite its relatively pleasant properties compared to the original Sobol-Hoeffding decomposition, nearly two decades after its introduction it has received relatively little attention when compared with the Sobol-Hoeffding alternative defined by conditional projections. (It should be briefly noted that in the case of independent variables, the same solution is recovered.) Amongst its limitations, beyond a lack of intuitive meaning behind its prescribed functions, the most severe is seemingly its computational intractability. It is rare to see a calculation of the full decomposition beyond a small number of dimensions or for distributions which are not piecewise constant. Unlike the conditional projection which can be more efficiently approximated from the bottom-up, the Hooker decomposition seems to endure the full exponential complexity of constructing a functional decomposition from the top-down (starting with the most complex $\tilde{h}_{[d]}$.)

Practical approaches to providing a solution to the full Hooker problem imitate Sinkhorn approximations via iterative refinement across the different variable axes (Lengerich et al., 2020). Nonetheless, the Sinkhorn algorithm has itself escaped a general closed form solution for decades (Sinkhorn, 1967; Nathanson, 2019), and practical application of the original Hooker decomposition has remained extremely limited.

In the space of Generalized Additive Models, however, recently some progress has been made. By leveraging the GAM's ability to reduce the exponential complexity of the true function to a lower-dimensional representation, works like Lengerich et al. (2020) and Sun et al. (2022) have found success in purifying the terms of an additive models. However, this success is still limited to two-or three- dimensional GAM models, being limited by: discrete variables assumptions requiring histogramming or kerneling of continuous variables; difficult transformations which do not easily scale to higher orders; and/or the previously discussed Sinkhorn-like approximations without clear guarantees. We later revisit these considerations more thoroughly in Appendix E after the introduction of our novel results for additive models, suggesting how our variational perspective potentially allows to unlock the same advantages for Hooker-type purified models.

**Further Alternatives** We highly recommend the work of Covert et al. (2021) for a very comprehensive review of potential methods for removal; however, we quickly review some of the major flavors for the counterfactual 'removal' of a feature. Some of the important methods yet unmentioned include the utilization of surrogate models to explicitly or implicitly mask out the features. This can be done implicitly via the training of a masked surrogate predictor using the projection equations for mean-squared error or for KL divergence as used within this work (Covert et al., 2021; Jethani et al., 2022; Covert et al., 2023). There have also been pursuits through a more explicit approach via using a separate generative model (VAE or GAN) as a proxy for removal (Chang et al., 2019), additionally allowing for more domain-specific approaches like image blurring and infilling. Another important set of alternatives is via the language of causality as introduced via Janzing et al. (2020). Unfortunately, after the introduction of elegant causal notation, the authors immediately use a simplifying assumption to reduce to the marginal Shapley, which has the aforementioned problems, only considering the engineer-level causality of 'causing the model' to change its predictions. This is significantly different from the scientist-level causality of 'causing the output' and has only begun to be thoroughly addressed in recent works like Biparva & Materassi (2024).

## A.2 POST-HOC FEATURE ATTRIBUTION AND INTERACTION ATTRIBUTION

**Notation** Let $d \in \mathbb{N}$ and $c \in \mathbb{N}$ be the dimensions of the input and output spaces, $\mathcal{X} \subseteq \mathbb{R}^d$ and $\mathcal{Y} \subseteq \mathbb{R}^c$. Let $F : \mathcal{X} \to \mathcal{Y}$ be a function representing a machine learning model which maps from inputs to outputs. We will use $[d] := \{1, \dots, d\}$ to represent the set of input features and $S \subseteq [d]$ to represent a subset of the input features. We also write the set of all such subsets, the powerset, as $\mathcal{P}([d]) \cong \{0, 1\}^d$ and use slight abuse of notations including $(S+i) := S \cup \{i\}$ and $(S-i) := S \setminus \{i\}$.

We will write the function space as some $\mathcal{H} = \{F : \mathcal{X} \to \mathcal{Y}\}$ and a masked function space as $\mathcal{H}' = \{f : \mathcal{X} \times \mathcal{P}([d]) \to \mathbb{R}\}$. For a general feature attribution method, we write $\Phi : \mathcal{H} \to \mathcal{H}^d$, taking a function $F(x)$ as input and returning a local explanation function $[\Phi_i \circ F](x)$ for each feature $i \in [d]$ on each local input $x \in \mathcal{X}$. Similarly, we define a blackbox feature attribution method as $\phi : \mathcal{H}' \to \mathcal{H}^d$, instead taking a masked function $f(x, S)$ as input and returning a local explanation function for each feature, $[\phi_i \circ f](x)$.

In addition to the notation introduced in the main body, we introduce some notation which are very useful in the domain of feature interactions. We first define the discrete derivative operator:

$$[\delta_i \circ f](T) = f(T + i) - f(T - i), \tag{24}$$

and its higher-order counterpart:

$$[\delta_S \circ f](T) := \sum_{W \subseteq S} (-1)^{|S|-|W|} f(T - S + W). \tag{25}$$

We note that the decision to add and remove elements instead of only adding elements is not necessarily typical; however, we find it beneficial to not need to restrict the domain of the discrete derivative operator.

We may now define the Mobius transformation or purification transformation as the one which replaces each function with its purified version, $\mu : \mathcal{H}' \to \mathcal{H}'$.

$$[\mu \circ f](x, S) := \tilde{f}(x, S) = \sum_{W \subseteq S} (-1)^{|S|-|W|} f(x, W) \tag{26}$$

We can also see that the purified functions can additionally be written in terms of the discrete derivative operator.

$$\tilde{f}(x, S) = [\delta_S \circ f](x, \emptyset) = \sum_{W \subseteq S} (-1)^{|S|-|W|} f(x, \emptyset + W) \tag{27}$$

Both the discrete derivative operator and the Mobius purification transformation are important tools for being able to more easily study the case of feature interactions in blackbox explainers. In the sections that follow we will define the major feature attribution and feature interaction attribution methods in terms of these operators.

**The Shapley Value**   The most typical definition of the Shapley value is usually its closed form solution as the weighted average of 1D derivatives,

$$[\phi^{\mathrm{SHAP}} \circ f]_i(x) = \sum_{S \subseteq [d]-i} \left[ \frac{1}{d} \binom{d-1}{|S|}^{-1} \cdot [\delta_i f](x, S) \right],$$

although its definition as an expectation over random permutations,

$$[\phi_i^{\mathrm{SHAP}} \circ f](x) = \frac{1}{|\mathcal{S}_d|} \sum_{\pi \in \mathcal{S}_d} \left[ [\delta_i f](x, S_{\pi,i}) \right],$$

has gained popularity in practice due to its susceptibility to Monte-Carlo sampling. We define $\mathcal{S}_d$ as the symmetric group or set of permutations on $d$ elements, $\mathcal{S}_d := \{\pi : [d] \to [d] \text{ s.t. } \pi \text{ is bijective}\}$, and we define $S_{\pi,i}$ as the set of predecessors to $i$ under the ordering $\pi$, $S_{\pi,i} := \{j \in [d] \text{ s.t. } \pi(j) < \pi(i)\}$.

We also state the alternative formulation in terms of 'unanimity games' which the authors believe to yield a more intuitive understanding of the Shapley value.

$$[\phi^{\mathrm{Sh}} \circ f]_i(x) = \sum_{S \supseteq \{i\}} \frac{\tilde{f}_S(x_S)}{|S|} \tag{28}$$

In words, the Shapley value divides the purified interaction $\tilde{f}_S(x_S)$ (which is the value created by $S$ and only $S$) amongst all of its constituent features, $i \in S$, completely uniformly between them, $\frac{1}{|S|}$.

We rewrite the four Shapley axioms in terms of the functional notation:

1. **Dummy**    If $[\delta_i \circ f](x, S) = 0$ for all $S$,
   then $[\phi_i \circ f](x) = 0$ (for that local $x$).
2. **Symmetry**    $\pi^{-1} \circ \phi_{\pi(i)} \circ \pi \circ f = \phi_i \circ f$    $\forall i \in [d], \forall \pi \in \mathcal{S}_d$
3. **Efficiency**    $\sum_{i \in [d]} \phi_i \circ f = f_{[d]} - f_\emptyset$
4. **Linearity**    $\phi \circ (f + g) = \phi \circ f + \phi \circ g$

It should be emphasized that dummy is a truly local property whereas symmetry, efficiency, and linearity can all be realized as properties of the additive functions. Thus, from the functional perspective, it is more appropriate to call this property 'local dummy' to emphasize its distinction from 'global dummy' functions which would be the case when $\phi_i \circ f \equiv 0$. We hope this would help eliminate the common confusion we discuss later in Appendix B. Another point to briefly note is that linearity condition is about the linearity of the operator rather than the linearity of the function.

**Other Common Explainers**   One of the original blackbox explainers is the LIME value (Ribeiro et al., 2016), which can also be written in this functional notation as:

$$[\Phi_i^{\mathrm{LIME}} \circ F](x) := \operatorname*{arg\,min}_{\phi \in \mathbb{R}^d} \left\{ \mathbb{E}_{S \sim p^{\mathrm{LIME}}(S)} \left[ \left| f^{\mathrm{LIME}}(x, S) - \sum_{i=1}^{d} \mathbb{1}(i \in S) \cdot \phi_i \right|^2 \right] \right\} \tag{29}$$

where the distribution is taken over a LIME distribution $p^{\text{LIME}}(S)$, and the function $f^{\text{LIME}}(x, S)$ is taken as the semi-local average value according to a data-dependent LIME kernel which is the exponential of some distance function. For further details see Ribeiro et al. (2016) or Lundberg & Lee (2017).

Another common set of explainers are the extremely simple 'leave-one-in' and 'leave-one-out' values, based on including a single feature or removing a single feature:

$$[\phi_i^{\text{inc}} \circ f](x) := [\delta_i \circ f](x, \emptyset) \tag{30}$$

$$[\phi_i^{\text{rem}} \circ f](x) := [\delta_i \circ f](x, [d]) \tag{31}$$

These also have equivalent versions for measuring the interaction effect in the more general 'inclusion value' or 'removal value':

$$[\phi_S^{\text{inc}} \circ f](x) := [\delta_S \circ f](x, \emptyset) \tag{32}$$

$$[\phi_S^{\text{rem}} \circ f](x) := [\delta_S \circ f](x, [d]) \tag{33}$$

We note that it is also popular to refer to the difference, $f(x, S) - f(x, \emptyset)$, rather than the interaction effect, as the inclusion value. Another important interaction explainer is the Archipelago value (Tsang et al., 2020) which is defined to be the average of these two $\phi_S^{\text{arch}} := \frac{1}{2}\phi_S^{\text{inc}} + \frac{1}{2}\phi_S^{\text{rem}}$. This simple estimator is surprisingly robust at detecting feature interactions that $\phi_S^{\text{inc}}$ or $\phi_S^{\text{rem}}$ would each individually miss.

There are also a few other game-theoretic approaches which have attracted attention recently such as the Banzhaf value (Banzhaf, 1965; Tsai et al., 2023; Wang & Jia, 2023; Enouen et al., 2024) and the Deegan-Packel index (Deegan & Packel, 1978; Biradar et al., 2024), especially in their application to classification tasks instead of regression tasks.

### A.3 Shapley Interaction Indices

**Shapley Interaction Indices** The first definition extending the Shapley value to try handling feature interactions was already constructed in 1999, mainly by the removal of the efficiency axiom (Grabisch & Roubens, 1999). This allows for a relatively simple extension using the permutation symmetry axiom to define the interaction index as a random order value where both features must be present rather than the one. From Table 1 below, it can be seen how this index divides the higher-order interaction effects amongst their constituent lower-order subsets in the same way as the original Shapley value ($1/t$).

Table 1: Shapley Interaction Indices for $k = 1, 2, 3$

|  |  | Equation | $t = 1$ | $t = 2$ | $t = 3$ | 4 | 5 | 6 | 7 | 8 | 9 | 10 |
|---|---|---|---|---|---|---|---|---|---|---|---|---|
| $k = 1$ | $s = 1$ | $1/t$ | 1 | $1/2$ | $1/3$ | $1/4$ | $1/5$ | $1/6$ | $1/7$ | $1/8$ | $1/9$ | $1/10$ |
| $k = 2$ | $s = 2$ | $1/(t-1)$ | 0 | 1 | $1/2$ | $1/3$ | $1/4$ | $1/5$ | $1/6$ | $1/7$ | $1/8$ | $1/9$ |
| $k = 3$ | $s = 3$ | $1/(t-2)$ | 0 | 0 | 1 | $1/2$ | $1/3$ | $1/4$ | $1/5$ | $1/6$ | $1/7$ | $1/8$ |

|  |  | Equation | $t = 11$ | $t = 12$ | $t = 13$ | 14 | 15 | 16 | 17 | 18 | 19 | 20 |
|---|---|---|---|---|---|---|---|---|---|---|---|---|
| $k = 1$ | $s = 1$ | $1/t$ | $1/11$ | $1/12$ | $1/13$ | $1/14$ | $1/15$ | $1/16$ | $1/17$ | $1/18$ | $1/19$ | $1/20$ |
| $k = 2$ | $s = 2$ | $1/(t-1)$ | $1/10$ | $1/11$ | $1/12$ | $1/13$ | $1/14$ | $1/15$ | $1/16$ | $1/17$ | $1/18$ | $1/19$ |
| $k = 3$ | $s = 3$ | $1/(t-2)$ | $1/9$ | $1/10$ | $1/11$ | $1/12$ | $1/13$ | $1/14$ | $1/15$ | $1/16$ | $1/17$ | $1/18$ |

**Shapley-Taylor Interaction Indices**  The next major advancement to Shapley interaction indices came with the introduction of the Shapley-Taylor indices in 2019 (Sundararajan et al., 2020). These interaction indices reintroduce the efficiency condition in a way which we now know reflects the additive model structure of summing to the full prediction. However, they achieve this decomposition by treating the lower-order additive effects asymmetrically from the maximum rank effects. In particular, they zero out the influence of everything except the purified effect and distribute the higher-order effects amongst the rank $k$ subsets. This can be seen more clearly in Tables 2, 3, and 4.

Table 2: Shapley-Taylor Coefficients for $k = 1$

| | | Equation | $t = 1$ | $t = 2$ | $t = 3$ | 4 | 5 | 6 | 7 | 8 | 9 | 10 |
|---|---|---|---|---|---|---|---|---|---|---|---|---|
| $k = 1$ | $s = 1$ | $1/t$ | 1 | $1/2$ | $1/3$ | $1/4$ | $1/5$ | $1/6$ | $1/7$ | $1/8$ | $1/9$ | $1/10$ |

| | | Equation | $t = 11$ | $t = 12$ | $t = 13$ | 14 | 15 | 16 | 17 | 18 | 19 | 20 |
|---|---|---|---|---|---|---|---|---|---|---|---|---|
| $k = 1$ | $s = 1$ | $1/t$ | $1/11$ | $1/12$ | $1/13$ | $1/14$ | $1/15$ | $1/16$ | $1/17$ | $1/18$ | $1/19$ | $1/20$ |

Table 3: Shapley-Taylor Coefficients for $k = 2$

| | | Equation | $t = 1$ | $t = 2$ | $t = 3$ | 4 | 5 | 6 | 7 | 8 | 9 | 10 |
|---|---|---|---|---|---|---|---|---|---|---|---|---|
| $k = 2$ | $s = 1$ | | 1 | 0 | 0 | 0 | 0 | 0 | 0 | 0 | 0 | 0 |
| | $s = 2$ | | 0 | 1 | $1/3$ | $1/6$ | $1/10$ | $1/15$ | $1/21$ | $1/28$ | $1/36$ | $1/45$ |

| | | Equation | $t = 11$ | $t = 12$ | $t = 13$ | 14 | 15 | 16 | 17 | 18 | 19 | 20 |
|---|---|---|---|---|---|---|---|---|---|---|---|---|
| $k = 2$ | $s = 1$ | | 0 | 0 | 0 | 0 | 0 | 0 | 0 | 0 | 0 | 0 |
| | $s = 2$ | | $1/55$ | $1/66$ | $1/78$ | $1/91$ | $1/105$ | $1/120$ | $1/136$ | $1/153$ | $1/171$ | $1/190$ |

Table 4: Shapley-Taylor Coefficients for $k = 3$

| | | Equation | $t = 1$ | $t = 2$ | $t = 3$ | 4 | 5 | 6 | 7 | 8 | 9 | 10 |
|---|---|---|---|---|---|---|---|---|---|---|---|---|
| $k = 3$ | $s = 1$ | | 1 | 0 | 0 | 0 | 0 | 0 | 0 | 0 | 0 | 0 |
| | $s = 2$ | | 0 | 1 | 0 | 0 | 0 | 0 | 0 | 0 | 0 | 0 |
| | $s = 3$ | | 0 | 0 | 1 | $1/4$ | $1/10$ | $1/20$ | $1/35$ | $1/56$ | $1/84$ | $1/120$ |

| | | Equation | $t = 11$ | $t = 12$ | $t = 13$ | 14 | 15 | 16 | 17 | 18 | 19 | 20 |
|---|---|---|---|---|---|---|---|---|---|---|---|---|
| $k = 3$ | $s = 1$ | | 0 | 0 | 0 | 0 | 0 | 0 | 0 | 0 | 0 | 0 |
| | $s = 2$ | | 0 | 0 | 0 | 0 | 0 | 0 | 0 | 0 | 0 | 0 |
| | $s = 3$ | | $1/165$ | $1/220$ | $1/286$ | $1/364$ | $1/455$ | $1/560$ | $1/680$ | $1/816$ | $1/969$ | $1/1140$ |

**n-Shapley Values**    The n-Shapley values are a more recent attempt to revitalize the original Shapley interaction indices to obey the efficiency axiom in an alternate way (Bordt & von Luxburg, 2023). They use a recursive form so that the maximum rank terms ($s = k$) are the same as the original interaction index (Grabisch & Roubens, 1999); however, the lower order terms ($s < k$) are chosen to exactly obey the efficiency terms. This requires the use of the Beroulli numbers to balance these terms in a recursive expansion. The first few orders can be seen in Tables 5, 6, and 7. Note the similarities and differences with Table 1.

Table 5: n-Shapley Coefficients for $n = k = 1$

| | | Equation | $t = 1$ | $t = 2$ | $t = 3$ | 4 | 5 | 6 | 7 | 8 | 9 | 10 |
|---|---|---|---|---|---|---|---|---|---|---|---|---|
| $k = 1$ | $s = 1$ | $1/t$ | 1 | $1/2$ | $1/3$ | $1/4$ | $1/5$ | $1/6$ | $1/7$ | $1/8$ | $1/9$ | $1/10$ |

| | | Equation | $t = 11$ | $t = 12$ | $t = 13$ | 14 | 15 | 16 | 17 | 18 | 19 | 20 |
|---|---|---|---|---|---|---|---|---|---|---|---|---|
| $k = 1$ | $s = 1$ | $1/t$ | $1/11$ | $1/12$ | $1/13$ | $1/14$ | $1/15$ | $1/16$ | $1/17$ | $1/18$ | $1/19$ | $1/20$ |

Table 6: n-Shapley Coefficients for $n = k = 2$

| | | Equation | $t = 1$ | $t = 2$ | $t = 3$ | 4 | 5 | 6 | 7 | 8 | 9 | 10 |
|---|---|---|---|---|---|---|---|---|---|---|---|---|
| $k = 2$ | $s = 1$ | $\frac{-(t-2)}{2t}$ | 1 | 0 | $-1/6$ | $-1/4$ | $-3/10$ | $-1/3$ | $-5/14$ | $-3/8$ | $-7/18$ | $-2/5$ |
| | $s = 2$ | $\frac{1}{(t-1)}$ | 0 | 1 | $1/2$ | $1/3$ | $1/4$ | $1/5$ | $1/6$ | $1/7$ | $1/8$ | $1/9$ |

| | | Equation | $t = 11$ | $t = 12$ | $t = 13$ | 14 | 15 | 16 | 17 | 18 | 19 | 20 |
|---|---|---|---|---|---|---|---|---|---|---|---|---|
| $k = 2$ | $s = 1$ | $\frac{-(t-2)}{2t}$ | $-9/22$ | $-5/12$ | $-11/26$ | $-3/7$ | $-13/30$ | $-7/16$ | $-15/34$ | $-4/9$ | $-17/38$ | $-9/20$ |
| | $s = 2$ | $\frac{1}{(t-1)}$ | $1/10$ | $1/11$ | $1/12$ | $1/13$ | $1/14$ | $1/15$ | $1/16$ | $1/17$ | $1/18$ | $1/19$ |

Table 7: n-Shapley Coefficients for $n = k = 3$

| | | Equation | $t = 1$ | $t = 2$ | $t = 3$ | 4 | 5 | 6 | 7 | 8 | 9 | 10 |
|---|---|---|---|---|---|---|---|---|---|---|---|---|
| $k = 3$ | $s = 1$ | $\frac{(t-3)(t-4)}{12t}$ | 1 | 0 | 0 | 0 | $1/30$ | $1/12$ | $1/7$ | $5/24$ | $5/18$ | $7/20$ |
| | $s = 2$ | $\frac{-(t-3)}{2(t-1)}$ | 0 | 1 | 0 | $-1/6$ | $-1/4$ | $-3/10$ | $-1/3$ | $-5/14$ | $-3/8$ | $-7/18$ |
| | $s = 3$ | $\frac{1}{(t-2)}$ | 0 | 0 | 1 | $1/2$ | $1/3$ | $1/4$ | $1/5$ | $1/6$ | $1/7$ | $1/8$ |

| | | Equation | $t = 11$ | $t = 12$ | $t = 13$ | 14 | 15 | 16 | 17 | 18 | 19 | 20 |
|---|---|---|---|---|---|---|---|---|---|---|---|---|
| $k = 3$ | $s = 1$ | $\frac{(t-3)(t-4)}{12t}$ | $14/33$ | $1/2$ | $15/26$ | $55/84$ | $11/15$ | $13/16$ | $91/102$ | $35/36$ | $20/19$ | $17/15$ |
| | $s = 2$ | $\frac{-(t-3)}{2(t-1)}$ | $-2/5$ | $-9/22$ | $-5/12$ | $-11/26$ | $-3/7$ | $-13/30$ | $-7/16$ | $-15/34$ | $-4/9$ | $-17/38$ |
| | $s = 3$ | $\frac{1}{(t-2)}$ | $1/9$ | $1/10$ | $1/11$ | $1/12$ | $1/13$ | $1/14$ | $1/15$ | $1/16$ | $1/17$ | $1/18$ |

**Faith SHAP, the Faithful Shapley Index**    The other most recent attempt at a Shapley interaction index is the faithful Shapley interaction index. Instead of leveraging the permutation sampling symmetry of the original Shapley value, this work instead extends the Shapley value by means of its least-squares characterization. As we discuss extensively in this work, this can be seen as further utilizing the characteriztaion of Shapley values as an additive model approximation.

Tsai et al. (2023)'s Equation (16) solves for the faithful Shapey interaction indices from the perspective of the Mobius/purified functions:

$$\phi_S^{\text{Faith-SHAP-k}} \circ f = \tilde{f}_S + (-1)^{k-|S|} \frac{|S|}{k+|S|} \binom{k}{|S|} \sum_{T \supsetneq S, |T| > k} \frac{\binom{|T|-1}{k}}{\binom{|T|+k-1}{k+|S|}} \tilde{f}_T \tag{34}$$

In Tables 8, 9, and 10 below, we display what these coefficients are for the Mobius purified interaction effects. We additionally calculate a slightly simpler form of these coefficients in the claim below.

Table 8: FaithSHAP Coefficients for $k = 1$

| | | Equation | $t=1$ | $t=2$ | $t=3$ | 4 | 5 | 6 | 7 | 8 | 9 | 10 |
|---|---|---|---|---|---|---|---|---|---|---|---|---|
| $k=1$ | $s=1$ | $1/t$ | 1 | $1/2$ | $1/3$ | $1/4$ | $1/5$ | $1/6$ | $1/7$ | $1/8$ | $1/9$ | $1/10$ |

| | | Equation | $t=11$ | $t=12$ | $t=13$ | 14 | 15 | 16 | 17 | 18 | 19 | 20 |
|---|---|---|---|---|---|---|---|---|---|---|---|---|
| $k=1$ | $s=1$ | $1/t$ | $1/11$ | $1/12$ | $1/13$ | $1/14$ | $1/15$ | $1/16$ | $1/17$ | $1/18$ | $1/19$ | $1/20$ |

Table 9: FaithSHAP Coefficients for $k = 2$

| | | Equation | $t=1$ | $t=2$ | $t=3$ | 4 | 5 | 6 | 7 | 8 | 9 | 10 |
|---|---|---|---|---|---|---|---|---|---|---|---|---|
| $k=2$ | $s=1$ | $\frac{-2(t-2)}{t(t+1)}$ | 1 | 0 | $-1/6$ | $-1/5$ | $-1/5$ | $-4/21$ | $-5/28$ | $-1/6$ | $-7/45$ | $-8/55$ |
| | $s=2$ | $\frac{6}{t(t+1)}$ | 0 | 1 | $1/2$ | $3/10$ | $1/5$ | $1/7$ | $3/28$ | $1/12$ | $1/15$ | $3/55$ |

| | | Equation | $t=11$ | $t=12$ | $t=13$ | 14 | 15 | 16 | 17 | 18 | 19 | 20 |
|---|---|---|---|---|---|---|---|---|---|---|---|---|
| $k=2$ | $s=1$ | $\frac{-2(t-2)}{t(t+1)}$ | $-3/22$ | $-5/39$ | $-11/91$ | $-4/35$ | $-13/120$ | $-7/68$ | $-5/51$ | $-16/171$ | $-17/190$ | $-3/35$ |
| | $s=2$ | $\frac{6}{t(t+1)}$ | $1/22$ | $1/26$ | $3/91$ | $1/35$ | $1/40$ | $3/136$ | $1/51$ | $1/57$ | $3/190$ | $1/70$ |

Table 10: FaithSHAP Coefficients for $k = 3$

| | | Equation | $t=1$ | $t=2$ | $t=3$ | 4 | 5 | 6 | 7 | 8 | 9 | 10 |
|---|---|---|---|---|---|---|---|---|---|---|---|---|
| $k=3$ | $s=1$ | $\frac{3(t-3)(t-2)}{t(t+1)(t+2)}$ | 1 | 0 | 0 | $1/20$ | $3/35$ | $3/28$ | $5/42$ | $1/8$ | $7/55$ | $7/55$ |
| | $s=2$ | $\frac{24(t-3)}{t(t+1)(t+2)}$ | 0 | 1 | 0 | $-1/5$ | $-8/35$ | $-3/14$ | $-4/21$ | $-1/6$ | $-8/55$ | $-7/55$ |
| | $s=3$ | $\frac{60}{t(t+1)(t+2)}$ | 0 | 0 | 1 | $1/2$ | $2/7$ | $5/28$ | $5/42$ | $1/12$ | $2/33$ | $1/22$ |

| | | Equation | $t=11$ | $t=12$ | $t=13$ | 14 | 15 | 16 | 17 | 18 | 19 | 20 |
|---|---|---|---|---|---|---|---|---|---|---|---|---|
| $k=3$ | $s=1$ | $\frac{3(t-3)(t-2)}{t(t+1)(t+2)}$ | $18/143$ | $45/364$ | $11/91$ | $33/280$ | $39/340$ | $91/816$ | $35/323$ | $2/19$ | $68/665$ | $153/1540$ |
| | $s=2$ | $\frac{24(t-3)}{t(t+1)(t+2)}$ | $-16/143$ | $-9/91$ | $-8/91$ | $-11/140$ | $-6/85$ | $-13/204$ | $-56/969$ | $-1/19$ | $-32/665$ | $-17/385$ |
| | $s=3$ | $\frac{60}{t(t+1)(t+2)}$ | $5/143$ | $5/182$ | $2/91$ | $1/56$ | $1/68$ | $5/408$ | $10/969$ | $1/114$ | $1/133$ | $1/154$ |

**Claim 1.**  The Faithful Shapley coefficients can be written with the alternative formula:

$$\phi_S^{\text{Faith-SHAP-k}} \circ f = \tilde{f}_S + \sum_{T \supsetneq S, |T| > k} (-1)^{k-s} \binom{k+s-1}{s-1} \frac{\binom{t-s-1}{k-s}}{\binom{t+k-1}{k}} \tilde{f}_T \tag{35}$$

*Proof.*  We would like to show that:

$$(-1)^{k-s} \frac{s}{k+s} \binom{k}{s} \frac{\binom{t-1}{k}}{\binom{t+k-1}{k+s}} = (-1)^{k-s} \binom{k+s-1}{s-1} \frac{\binom{t-s-1}{k-s}}{\binom{t+k-1}{k}}$$

Let us write

$$\left[ \frac{s}{k+s} \binom{k}{s} \binom{t-1}{k} \binom{t+k-1}{k+s}^{-1} \right] \cdot \left[ \binom{k+s-1}{s-1} \binom{t-s-1}{k-s} \binom{t+k-1}{k}^{-1} \right]^{-1} =$$

$$\left[\frac{s}{k+s}\cdot\frac{k!}{s!(k-s)!}\frac{(t-1)!}{k!(t-k-1)!}\binom{t+k-1}{k+s}^{-1}\right]\cdot\left[\frac{(s-1)!k!}{(k+s-1)!}\binom{t-s-1}{k-s}^{-1}\frac{(t+k-1)!}{k!(t-1)!}\right]=$$

$$\left[\frac{1}{k+s}\cdot\frac{(t-1)!}{(s-1)!(k-s)!(t-k-1)!}\binom{t+k-1}{k+s}^{-1}\right]\cdot\left[\frac{(s-1)!(t+k-1)!}{(k+s-1)!(t-1)!}\binom{t-s-1}{k-s}^{-1}\right]=$$

$$\left[\frac{1}{k+s}\cdot\frac{1}{(k-s)!(t-k-1)!}\binom{t+k-1}{k+s}^{-1}\right]\cdot\left[\frac{(t+k-1)!}{(k+s-1)!}\binom{t-s-1}{k-s}^{-1}\right]=$$

$$\left[\frac{1}{k+s}\cdot\frac{1}{(k-s)!(t-k-1)!}\frac{(t-s-1)!(k+s)!}{(t+k-1)!}\right]\cdot\left[\frac{(t+k-1)!}{(k+s-1)!}\frac{(k-s)!(t-k-1)!}{(t-s-1)!}\right]=$$

$$\left[\frac{1}{k+s}\cdot\frac{(k+s)!}{(t+k-1)!}\right]\cdot\left[\frac{(t+k-1)!}{(k+s-1)!}\right]=$$

$$\left[\frac{(k+s-1)!}{(t+k-1)!}\right]\cdot\left[\frac{(t+k-1)!}{(k+s-1)!}\right]=$$

$$\left[1\right]\cdot\left[1\right]=1$$

$\square$

Due to the fact that $s \leq k < t$, we find this to be a slightly nicer formulation when written as a function of $t$.

## B COMMON PITFALLS IN THE CURRENT LITERATURE

### B.1 LOCALLY ZERO VS. GLOBALLY ZERO

Consider the simple function $f(x_1, x_2) = \cos(2\pi x_1) + \sin(2\pi x_2)$. Assume the input features are uniformly distributed across the space $[-1, 1]^2$. Based on our correspondence, we can see see that the Shapley value functions decompose the same as an additive model with $\phi_1(x_1, x_2) = \cos(2\pi x_1)$ and $\phi_2(x_1, x_2) = \sin(2\pi x_2)$.

When we set $x_1 = \frac{1}{4}$, we have that $\phi_1(\frac{1}{4}, x_2) = \cos\frac{\pi}{2} = 0$. Does this suddenly mean that our function $\phi_1(x_1, x_2)$ is not a function of $x_1$ because it is zero at one value? No, it does not.

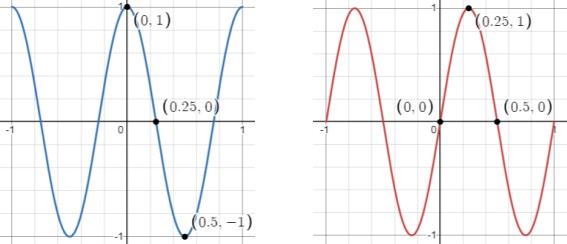

Figure 7: Cosine and sine functions.

It is hoped that after the clarification of the functional perspective on the Shapley value, it can be made clear that the exact same question is being asked when the Shapley value is equal to zero for a single input point. If one is interested in the global importance of a feature, then one should check for being zero or nonzero as an entire function. This can for instance be checked with the variance of the Shapley function:

$$\mathbb{V}_i^{\text{SHAP}} := \mathbb{V}\text{ar}_X[\phi_i^{\text{SHAP}}(X)] > 0 \tag{36}$$

### B.2 BEESWARM PLOTS INSTEAD OF SHAPE FUNCTIONS

It is common to see beeswarm plots of the SHAP values used as an aggregate summary over an entire dataset. Although it is an information dense representation of the SHAP values across the entire dataset, we feel the additive model perspective brings some insights into their limitations and what can be improved about them.

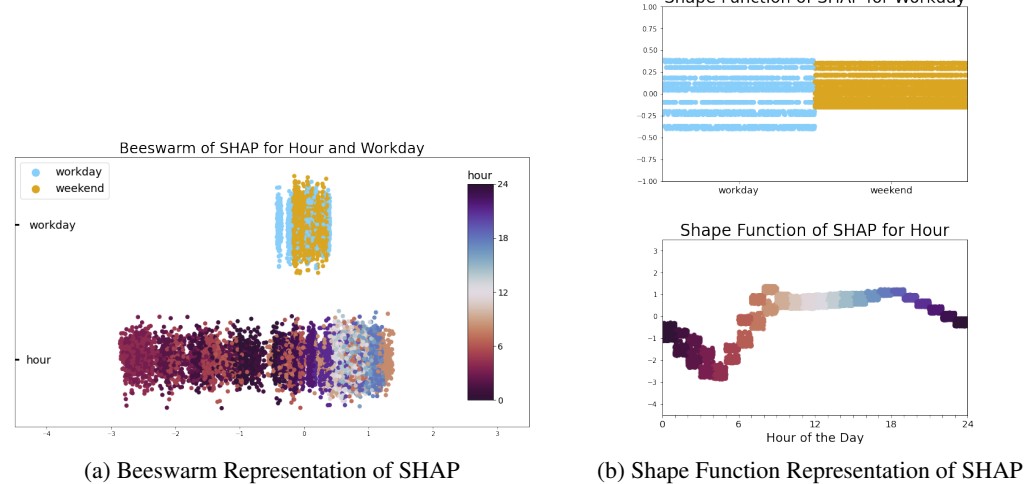

(a) Beeswarm Representation of SHAP      (b) Shape Function Representation of SHAP

Figure 8: Alternate representations of the SHAP values aggregated over an entire dataset.

First of all, the major limitation when compared to the shape function or additive model representation is simply the compression of information. Each of the shape functions in Figure 8b is com-

pressed horizontally and then its flattened version is rotated and stacked amongst the SHAP values for all other features. Although this information about the feature value is theoretically preserved via the colormap, it is well known from data visualization that this is insufficient to adequately preserve the information. This compression effect is especially pronounced in cyclic or heavily oscillating shape functions.

Surprisingly, these plots are also commonly used with the same red-blue diverging colormap which is used in the output space for SHAP. This colormap makes sense for the SHAP values since they are referring to the positive or negative influence on the output prediction; however, when also applied to the feature value, we are conflating the y-axis and the x-axis in the shape functions of Figure 8b. Additionally, this is using a diverging colormap to represent what are usually sequential features (meaning a sequential colormap should be used instead). Although we again recommend to look at the shape functions to get a clearer overall picture, if one is not willing to inspect all of the shape functions, it is perhaps more effective to remove the color from the beeswarm plot entirely and/ or to just use an aggregate statistic such as the variance in Equation 36 to measure the spread of the Shapley values depicted by the beeswarm plot.

### B.3 LOCALLY LINEAR OR LOCALLY ADDITIVE

The interpretation of a locally linear model has two key interpretations which are unfortunately conflated in much of the work on explainability. The first is the semi-local interpretation of the coefficients as gradient-like coefficients telling the direction of greatest influence. Usually this is done similar to LIME (Ribeiro et al., 2016) where a linear function is fit to a weighted neighborhood of points. This is in contrast with the gradient which fits the same linear function to an arbitrarily small neighborhood of points. These are both different from the second interpretation of the linear model as a structural assumption onto the functional space. In particular, the linearity and efficiency axioms of the Shapley value are made on the Shapley operator itself. The local linearity assumptions do not translate to a global linearity assumption, but rather to a global additive assumption where the height function is dictated to respect the additive structure of the original function.

Hopefully after the clearer connections we make with additive models, it is clear why the height interpretation is the correct one for SHAP and that the gradient-like tests for SHAP are ill-posed after a contextual understanding of the goal of SHAP as the height function rather than a measurement of the local sensitivity similar to the gradient.

Let us also briefly recall that in the case of a GAM, there is still an interesting correspondence between the gradient of SHAP and the gradient of the GAM.

Recall the GAM-1 equation.
$$F^{\leq 1}(x_1, \ldots, x_d) = f_1(x_1) + \cdots + f_d(x_d)$$
It follows that the gradient obeys the similar
$$\nabla F^{\leq 1}(x_1, \ldots, x_d) = \langle \partial_1 f_1(x_1), \ldots, \partial_d f_d(x_d) \rangle$$
$$= \langle \partial_1 \phi_1(x), \ldots, \partial_d \phi_d(x) \rangle$$

If one is interested in a local sensitivity test, then something like the gradient of the GAM or gradient of the SHAP should instead be used, but it is gently reminded that the vanilla gradient will ignore the statistical structure of the manifold (Frye et al., 2021) and could face alternate issues like the shattered gradients problem (Balduzzi et al., 2017).

### B.4 BASELINE METHOD ON DISCRETE INPUTS

If you have a finite set of discrete, categorical inputs, then the (mis)usage of the baseline method becomes of great importance for reasons beyond the off-the-manifold problem. In particular, it is common to replace the current input variable by the baseline input variable only to realize they had the same value (e.g. zero). It follows that the counterfactual removal will have no effect, Although extremely rare in the continuous case, when using categorical input variables it is easy to mask out bivariate and even higher-order effects. It is especially easy to make this mistake on boolean input variables. Accordingly, it is perhaps recommended to those with lesser familiarity with boolean functions or logical functions to use the $\{\pm 1\}$ or one-hot encoding instead of $\{0, 1\}$ to avoid making such mistakes (O'Donnell, 2014).

## C    IMPOSSIBILITY THEOREMS FOR FEATURE INTERACTIONS

This section will broadly prove the representational power of SHAP and related black-box explainers in the form of 'impossibility theorems' or 'possibility theorems'. The main method of proof technique is simply via the proof of functional correspondence between the two spaces, the functional space of interest to be taken for hypothesis tests, $\mathcal{H}$, and the functional space of additive models for some sufficiently large order $k \in \mathbb{N}$, $\mathcal{H}^{\leq k}$.

Let us first proceed by defining the trace of the SHAP function as the object of interest so that we may thus allow a direct functional comparison between the two of GAM and SHAP. For simplicity, we will assume a rectangular feature space $\mathcal{X} = \mathcal{X}_1 \times \cdots \times \mathcal{X}_d$ for some subspaces $\mathcal{X}_i$ for each $i \in [d]$. It is simple to extend some subset of $\mathbb{R}^d$ to a rectangular subset in this way by taking the product of the marginal spaces. It is also necessary to extend the distribution $p(x)$ which has been assumed on the original space $\mathcal{X}$ to our rectangular space. However, the obvious extension of zero probability on any additional coordinate will be sufficient for our purposes, since it will already be required to make all of our statements modulo differences on a null set (set of measure zero). Accordingly, we will simply restrict our focus to a rectangular feature space and ignore any concerns regarding differences on a null set. Any hypothesis test which is testing on a set of measure zero, although completely ill-posed, will not be answerable under this statistical framework.

For a given test point $x^* \in \mathcal{X}$, we define the SHAP trace at $x^*$ to be the object:

$$\mathcal{T}_F^{\text{SHAP}}(x^*) = \left\{ \left( x, [\Phi^{\text{SHAP}} \circ F](x) \right) : x \in \bigcup_{i \in [d]} \{x_1^*\} \times \cdots \times \{x_{i-1}^*\} \times \mathcal{X}_i \times \{x_{i+1}^*\} \times \cdots \times \{x_d^*\} \right\}$$

In words, the trace of the SHAP value is the image under the Shapley function of the set which includes all possible 1D perturbations of a single feature value.

To enable a direct comparison with the original functional space $\mathcal{H}$, we will define the completion of the trace as the function which is extended to the entire rectangular feature space $\mathcal{X}$ in the obvious way by assuming that the SHAP trace is the true additive model representing the underlying function $F$.

$$\mathcal{CT}_F^{\text{SHAP}}(x^*) = \left\{ \left( x, \sum_{i=1}^{d} [\Phi_i^{\text{SHAP}} \circ F](x_1^*, \ldots, x_{i-1}^*, x_i, x_{i+1}^*, \ldots, x_d^*) \right) : x \in \mathcal{X} \right\}$$

**Theorem 5.** The SHAP trace of a function $F$ will satisfy any hypothesis test $\mathcal{H}_0$ v. $\mathcal{H}_1$ inside of the functional space $\mathcal{H}$ if and only if the functional space $\mathcal{H}$ is equivalent to a shift which is not a superset of $\mathcal{H}_{\text{ANOVA}}^{\leq 1}$.

*Proof.* Since we are making a claim across all possible splittings of the functional space $\mathcal{H}$ into two possibilites of the null hypothesis $\mathcal{H}_0$ and the alternative hypothesis $\mathcal{H}_1$, it is required that we actually show the exact identifiability of the individual function from the SHAP trace alone.

Suppose first the $\mathcal{H}$ truly is a subset of the ANOVA-1 space (or a shift thereof). We simply take the difference $(F_1 - F_2)$ for some arbtirary $F_1, F_2 \in \mathcal{H}$ or otherwise assume we know the functional shift required to center our functional space to be a subset of the ANOVA-1 space. It follows from the exact formula of the SHAP function that for any $F \in \mathcal{H}$, we will have that $\phi_i(x) = \phi_i(x_i) = \tilde{f}_i(x_i)$, which implies that the completion of this trace will immediately recover the original GAM-1 function.

In the other direction, to prove the contrapositive, assume instead that $\mathcal{H}$ truly has some feature interaction, which can be represented by $(F_1 - F_2)$ and $(F_1 - F_3)$ being some different shifts for some $F_1, F_2, F_3 \in \mathcal{H}$. For simplicity, let us shift by $(F_1 - F_3)$ so that one difference is zero and one difference is nonzero. It follows that for the nonzero interaction effect, there is some $S \subseteq [d]$ with $|S| > 1$ such that $\tilde{f}_S \not\equiv 0$. Since this is true in the statistical sense, there is a region of sufficient difference. Via the assumption that our space $\mathcal{H}$ is at least as representative as $\mathcal{H}_{\text{ANOVA}}^{\leq 1}$, this means we can find two distinct functions which map to the same SHAP trace in the local region of this nonzero measure region. Accordingly, if we take a hypothesis test which identifies these two distinct functions, their SHAP traces will still look indistinguishable and there will be no succesful hypothesis test based on the SHAP trace. $\square$

Of course, practically speaking, we likely do not have access to a priori knowledge about the global feature interactions of some hypothesis space $\mathcal{H}$ which would allow for the construction of the isomporhism between our given $\mathcal{H}$ and some $\tilde{\mathcal{H}}$ which is actually a subset of the ANOVA-1 space $\mathcal{H}_{\text{ANOVA}}^{\leq 1}$

Proceeding by defining the trace of Faith-SHAP-k and GAM-k in the obvious way, we may find a similar theorem for the higher-order interactions of $k \in \mathbb{N}$. Once again, it is practically more useful to say we cannot directly assume the existence of feature interactions across the entire hypothesis space and hence again asssume $\mathcal{H} \subseteq \mathcal{H}_{\text{ANOVA}}^{\leq k}$ without the caveat of allowing a shift by some oracle assumption. It is also perhaps more interesting that we can make the same statement for any arbitrary frontier $\mathcal{I} \subseteq \mathcal{P}([d])$ as we will introduce in the general study of additive models in Appendix E.

### C.1 SPECIFIC EXAMPLES

Although we have shown exact functional equivalence from which the ability to do hypothesis tests follows, we restate some of the tests from Bilodeau et al. (2022) to make a clearer comparison.

The first hypothesis test is coming from their Proposition 3.5, and although they do not name this hypothesis test, we call it the "almost $\delta$-local Lipschitz" because of its resemblance to the slightly more typical "$\delta$-local Lipschitz" test.

$$\mathcal{L}_{\text{almost}}^{\delta,x,i}(F) := \sup_{x_i' \in [x_i - \delta, x_i + \delta]} \left\{ \frac{|F(x') - F(x)|}{\delta} \right\} \tag{37}$$

$$\mathcal{L}^{\delta,x,i}(F) := \sup_{x_i' \in [x_i - \delta, x_i + \delta]} \left\{ \frac{|F(x') - F(x)|}{|x' - x|} \right\} \tag{38}$$

They then create the hypothesis tests:

$$\mathcal{H}_0 = \left\{ F \in \mathcal{H} : \mathcal{L}_{\text{almost}}^{\delta,x,i}(F) \leq \frac{\varepsilon}{2} \right\} \tag{39}$$

$$\mathcal{H}_1 = \left\{ F \in \mathcal{H} : \mathcal{L}_{\text{almost}}^{\delta,x,i}(F) > \varepsilon \right\} \tag{40}$$

And then identify that the gradient can successfully distinguish these two hypotheses. This is desired since it is well known that on a compact interval, continuously differentiable functions are automatically Lipschitz functions.

The next two major hypothesis tests they build are for 'local recourse' in Definition 3.7 and 'locally spurious' in Definition 3.8. For recourse, we first assume some counterfactual distribution $\nu(x)$ which they implicitly assume to have nonzero measure across the left-local and right-local regions $[x_i - \delta, x_i]$ and $[x_i, x_i + \delta]$. We define the 'value of moving to the left' and the 'value of moving to the right' under the counterfactual distribution, $\nu(x)$, as the $\delta$-local left and right recourse values:

$$\mathcal{V}^{\delta,x,i,-}(F) := \mathbb{E}_{X \sim \nu} \left[ f(x_1, \ldots, X_i, \ldots, x_d) \mid X_i \in [x_i - \delta, x_i] \right] \tag{41}$$

$$\mathcal{V}^{\delta,x,i,+}(F) := \mathbb{E}_{X \sim \nu} \left[ f(x_1, \ldots, X_i, \ldots, x_d) \mid X_i \in [x_i, x_i + \delta] \right] \tag{42}$$

They then create the hypothesis tests:

$$\mathcal{H}_0 = \left\{ F \in \mathcal{H} : \mathcal{V}^{\delta,x,i,+}(F) > \mathcal{V}^{\delta,x,i,-}(F) \right\} \tag{43}$$

$$\mathcal{H}_1 = \left\{ F \in \mathcal{H} : \mathcal{V}^{\delta,x,i,+}(F) \leq \mathcal{V}^{\delta,x,i,-}(F) \right\} \tag{44}$$

The infinity norm over a local interval (from the left and from the right) is defined as:

$$\|F\|_{\infty}^{\delta,x,i,-} := \sup_{x_i' \in [x_i - \delta, x_i]} \left\{ \left| F(x') \right| \right\} \tag{45}$$

$$\|F\|_{\infty}^{\delta,x,i,+} := \sup_{x_i' \in [x_i, x_i + \delta]} \left\{ \left| F(x') \right| \right\} \tag{46}$$

The final major hypothesis tests introduced in that work are the tests for if a feature is local spurious:

$$\mathcal{H}_0 = \left\{ F \in \mathcal{H} : \|F\|_\infty^{\delta,x,i,+} = 0 \right\} \tag{47}$$

$$\mathcal{H}_1 = \left\{ F \in \mathcal{H} : \|F\|_\infty^{\delta,x,i,+} \geq \varepsilon \right\} \tag{48}$$

To reiterate, all of these tests are easily answered via the SHAP value using the additive model trace. All of these hypothesis tests focus on the behavior inside of a local neighborhood for some arbitrary $\delta$, whereas the SHAP value evaluated at a single point only describes the behavior at a single point. Hopefully, from the functional perspective on Shapley, it is now clear how easily all of these questions can be answered by using the additive model equivalent of the Shapley value. This does not conflict with previous works showing negative results on these same hypothesis test since their focus was on the ability of a pointwise indicator's ability to perform these hyptohesis tests, and is related to our discussion in Section B.1 and Section B.3. Moreover, it is hoped that the true limitation of SHAP, its inability to adequately handle feature interactions, is now emphasized as something that cannot be shown by any of these existing impossibility tests due to their focus on a single perturbed feature at a time.

## D EXPERIMENT DETAILS

### D.1 2D SYNTHETIC

First, we look at a particularly simple example of synthetic data to highlight the two important aspects which the Shapley value alone is unable to capture: feature interaction and feature correlation. Hopefully this example will help develop intuition for the Shapley value and highlight its unique challenges in the setting where input variables are correlated.

For some $\rho \in [-1, 1]$, we consider the data generated by

$$f(x, y) = x + xy \qquad X, Y \sim \mathcal{N}\left(\vec{0}, \begin{pmatrix} 1 & \rho \\ \rho & 1 \end{pmatrix}\right)$$

It is relatively straightforward to calculate that:

$$f_\emptyset = \rho \qquad\qquad \tilde{f}_\emptyset = \rho$$
$$f_x = x + \rho x^2 \qquad\qquad \tilde{f}_x = x + \rho x^2 - \rho$$
$$f_y = \rho y + \rho y^2 \qquad\qquad \tilde{f}_y = \rho y + \rho y^2 - \rho$$
$$f_{xy} = x + xy \qquad\qquad \tilde{f}_{xy} = -\rho y + xy - \rho x^2 - \rho y^2 + \rho$$

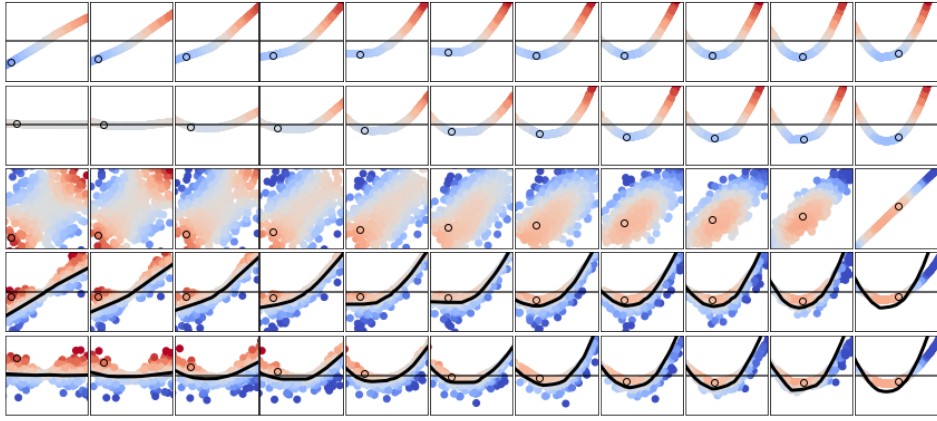

Figure 9: Simple Synthetic Dataset using various $\rho \in \{0.0, 0.1, 0.2, \dots, 1.0\}$. Each of the five rows corresponds to the learned $\tilde{f}_x$, $\tilde{f}_y$, $\tilde{f}_{xy}$, $\phi_x$, and $\phi_y$. The third row is hence 'top-down' from the z-axis whereas all others have the output f as the vertical axis. Visually constrained to $x, y \in [-2, 2]^2$ with colors/ outputs in $[-3, 3]$. A single point is highlighted to emphasize how the Shapley value in the bottom two rows is constructed from the top three rows.

The Sobol covariances are hence:

$$\mathbb{E}[f \cdot \tilde{f}_\emptyset] = \rho^2$$
$$\mathbb{E}[f \cdot \tilde{f}_x] = \mathbb{E}[x^2 + \rho x^3 + x^2 y + \rho x^3 y - \rho x - \rho xy] = 1 + 0 + 0 + 3\rho^2 + 0 - \rho^2$$
$$\mathbb{E}[f \cdot \tilde{f}_y] = \mathbb{E}[\rho xy + \rho xy^2 + \rho xy^2 + \rho xy^3 - \rho x - \rho xy] = \rho^2 + 0 + 0 + \rho(3\rho) + 0 - \rho^2$$
$$\mathbb{E}[f \cdot \tilde{f}_{xy}] = \mathbb{E}[f^2] - (\rho^2) - (1 + 2\rho^2) - (3\rho^2) = 1 + (1 + 2\rho^2) - (1 + 6\rho^2)$$

Hence $\mathbb{C}_\emptyset = \rho^2$, $\mathbb{C}_x = 1 + 2\rho^2$, $\mathbb{C}_y = 3\rho^2$, $\mathbb{C}_{xy} = 1 - 4\rho^2$.

The Shapley functions are also $\phi_x(x, y) = \tilde{f}_x + \frac{1}{2}\tilde{f}_{xy}$ and $\phi_y(x, y) = \tilde{f}_y + \frac{1}{2}\tilde{f}_{xy}$:

$$\phi_x(x, y) = (x + \rho x^2 - \rho) + \tfrac{1}{2}(-\rho y + xy - \rho x^2 - \rho y^2 + \rho) = \left[x - \tfrac{\rho}{2}y\right] + \left[\tfrac{xy}{2} + \tfrac{\rho}{2}(x^2 - y^2 - 1)\right]$$

$$\phi_y(x, y) = (\rho y + \rho y^2 - \rho) + \tfrac{1}{2}(-\rho y + xy - \rho x^2 - \rho y^2 + \rho) = \left[\tfrac{\rho}{2}y\right] + \left[\tfrac{xy}{2} + \tfrac{\rho}{2}(y^2 - x^2 - 1)\right]$$

In Figure 9, we can see the learned set of functions across various $\rho \in \{0.0, 0.1, \ldots, 1.0\}$ when learning a GAM with the purified loss function. The Shapley functions can also be calculated to be $\phi_x = \tilde{f}_x + \frac{1}{2}\tilde{f}_{xy}$ and $\phi_y = \tilde{f}_y + \frac{1}{2}\tilde{f}_{xy}$ which can be seen in the fourth and fifth rows from the Figure. We can furthermore see that the additive models using the purified loss align with the true purified ANOVA decomposition and that hence the Shapley value functions can be computed in constant time given the purified GAM model. Further, going left to right, we see how the strength of the 1D effects (first and second rows) increase, whereas the strength of the 2D effects (third row) decreases as the amount of correlation increases. This corresponds to the deterioration of the feature interaction due to the increase in feature correlation. At the halfway mark, the 2D function is no longer positively correlated with the true outcome. This is most obvious in the far right ($\rho = 1.0$) plots where $\tilde{f}_x = \hat{f}_x = -\tilde{f}_{xy}$, meaning $\phi_x = \frac{1}{2}\tilde{f}_x$ and $\phi_y = \frac{1}{2}\tilde{f}_y$.

Interestingly, we can see that for $|\rho| > \frac{1}{2}$, we actually have that $\mathbb{C}_{xy} < 0$. That is to say, the interaction term alone is no longer positively correlated with the function we are trying to learn. This further implies that the purified interaction is actually negatively correlated with our target, and adding it to the model somehow reduces the performance as measured with MSE. This must be juxtaposed with the fact that our function $f(x, y) = x + xy$ clearly demonstrates a feature interaction in the term $xy$.

This can however be resolved by not thinking of the purified interaction alone but in conjunction with other features when it is added to the model. For example, if we started with $y$ and added $x$, then we could consider $\mathbb{C}_x + \mathbb{C}_{xy} = 2 - 2\rho^2 \geq 0$ as the improvement to the model. Conversely, if we started with $x$ and added $y$, we could consider $\mathbb{C}_y + \mathbb{C}_{xy} = 1 - \rho^2 \geq 0$ as the improvement to the model. Intuitively, it is not the fact that the interaction is detrimental to the model performance, as clearly it is necessary for $|\rho| < 1$, but rather that it is overshadowed by the information which is gained from either $x$ or $y$ alone. "The redundant information from knowing $x$ or $y$ outweighs the synergistic information from knowing $x$ and $y$."

### D.2 10D Synthetic

For our major synthetic experiments where we benchmark the ability of FastSHAP and InstaSHAP to recover the Shapley value effects, we create a dataset similar to the simple ones in Figures 2 and 9. We generate ten features from a correlated pairs structure on the covariance matrix. In practice, this will mean our low-dimensional synthetic target variable will remain having a low-dimensional functional ANOVA decomposition because of this simplistic correlation structure. Moreover, this allows us to relatively easily calculate the exact Shapley functions even in this 10-dimensional dataset.

$$\Sigma = \begin{pmatrix} 1 & \rho & & & & & \\ \rho & 1 & & & & & \\ & & 1 & \rho & & & \\ & & \rho & 1 & & & \\ & & & & \ddots & & \\ & & & & & 1 & \rho \\ & & & & & \rho & 1 \end{pmatrix}$$

We then take the target variable to be

$$f(x) = \sum_{S \in \mathcal{I}^{\leq k*}} \beta_S \cdot \prod_{i \in S} x_i \tag{49}$$

for some $k* \in \mathbb{N}$ and for some $\beta_S$ drawn from the normal distribution $\mathcal{N}(0, 1)$ or the Laplace distribution $\text{Laplace}(0, 1)$. We finally divide by a constant to normalize the output response such that the total variance of the output is equal to one.

### D.3 Real-World Tabular Datasets

We follow the methods of SIAN (Enouen & Liu, 2022) to train GAM models for tabular datasets. After the training of a surrogate model, the Archipelago (Tsang et al., 2020) interaction detection method is applied to choose the most important feature interactions from the dataset. After a small

number of feature interactions are chosen from the dataset, a neural network which obeys this low-dimensional GAM structure is trained under the same loss function objective as in the main paper.

**Bikeshare** This dataset predicts the expected bike demand each hour given some relevant features like the day of the week, time of day, and current weather. There is a total of thirteen different input features predicting a single continuous output variable.

There is a gap in accuracy from GAM-1 to an MLP where the GAM-1 achieves an $R^2$ error of $17.4\%$, whereas an MLP achieves an $R^2$ error of $6.59\%$. Using the techniques of SIAN, we select 20 tuples of size three or less to train a GAM-3 model which achieves $6.23\%$ $R^2$ error, closing the gap between GAM-1≡SHAP through the usage of feature interactions.

In particular, it is well known that on this dataset there is a strong interaction between the hour variable and workday variable (since people's schedules change on the weekend vs. a workday.) The ability to capture this particular feature interaction is critical for accurately understanding the dataset, as seen in Figure 4. Also in Figure 4, it can be viewed how the interpretability-uninterpretability spectrum along the axis of additive models supports the hypothesis of Rudin (2019) by demonstrating that training an accurate GAM model is sufficient to explain SHAP; however, training an accurate SHAP score is not sufficient for training accurate GAM models.

**Treecover** The dataset consists of predicting the types of trees covering a specific forest area from a selection of 7 tree species (Spruce, Lodgepole Pine, Ponderosa Pine, Cottonwood, Aspen, Douglas-Fir, or Krummholz) in a Colorado national park based on 10 numerical features and 1 categorical feature of the area.

However, this misses the fact that both the elevation and soil type are additionally correlated with one another. Indeed, the soils are grouped according to climatic zone which generally correspond to different altitude climates. For convenience, we keep these soil classes in the same orders as their expected elevation, named: 'lower montane', 'upper montane', 'subalpine', and 'alpine'. One notes that the Krummholz tree can be found at high altitudes but also in alpine (often rocky) soil. Similarly, Cottonwoods, Douglas-firs, and Ponderosas are expected to be found at lower altitudes, but also to be found in montane soils. Without an understanding that these two features are correlated with one another, it might a priori seem like these are two independent contributions to the prediction. Yet again, it turns out that these two facts are indeed correlated with one another and hence the 1D projections alone may not be sufficient to yield a good explanation.

In this case, a lot can be gleaned by viewing the 2D shape function which depends on both the soil and the elevation. In Figure 5, we visualize this 2D shape function as a scatterplot with colored heatmap. Through the density of points, we can see there is indeed a strong positive correlation between the soil type and the elevation. Furthermore, using the colors for each tree species, we can see that there is a lot of redundant information carried by both the soil and the elevation, but also that there is some non redundant information.

We train an MLP on this dataset to achieve $80.4\%$ validation accuracy and we train a GAM-1 to achieve $72.4\%$ validation accuracy. This once again shows a gap in the feature interactions which discredits the ability of SHAP to provide an adequate explanation of what is being learned by the MLP model. Once again following the techniques of training lower-order GAMs, we are able to train a GAM-5 on 50 tuples to achieve $82.2\%$ accuracy. This shows that likely there is some information which the low-order GAM with interactions can understand that GAM-1 and SHAP are missing.

## D.4 Computer Vision

We perform experiments on the CUB dataset consisting of 200 different species of birds and containing over 6000 labeled images (Wah et al., 2011). In addition to the species level information, we construct a coarser-grained class label out of the taxonomic family of each of those bird species. This results in 37 coarse-grained labels for each bird. For our main CNN we train a ResNet-50 model (He et al., 2016) on the masked surrogate objective and for our GAM-$K$x$K$ architecture we train a modifed resnet to only allow for communication between adjacent patches of size 16x16, further details in code. Both models are initialized with mostly pretrained weights and fine-tuned for 300 epochs on the CUB dataset.

Our vanilla resnet is able to get to $65.0\%$ fine accuracy and $81.8\%$ coarse accuracy. Compare this to the $33.2\%$ fine and $53.7\%$ coarse accuracies of the GAM-1x1. There is clearly a sizable gap in performance between the full complexity ResNet and the GAM-restricted Resnet, indicating the importance of feature interactions and emphasizing the potential deceptiveness of SHAP on this dataset. The GAM-2x2 and GAM-3x3 achieve fine accuracies of $45.8\%$ and $46.8\%$ as well as coarse accuracies of $66.3\%$ and $66.8\%$. This once again indicates that even with some feature interactions, the performance of the Resnet is dependent on even higher order feature interactions or longer-range feature interactions. As discussed, it is not impossible that these conclusions are only true for the training method used in the GAM-$K$x$K$ and that some novel GAM architecture would not be able to achieve higher performance. Nonetheless, the issues in training modern neural network will be present also in alternative approaches like FastSHAP, implying that these conclusions are valid regardless.

Another major challenge of applying to domains like computer vision and natural language processing is the prolific usage of pretrained models for downstream tasks. As discussed in Covert et al. (2023), this brings up the important question of how to do surrogate-based modeling to compute the conditional expectation $\mathcal{M}$. In principle, we would like to also do the pretraining stage with surrogate masking, however, in practice previous works on this domain (Jethani et al., 2022; Covert et al., 2023) will instead use the pretrained models which are available and do the fine tuning stage with the masked objectives as we presented in the paper.

We briefly discuss the application of SHAP to classification and how it is different from its application to regression. In particular, it is often common to train on the cross-entropy or $D_{KL}$ objective, but still use the SHAP for regression directly on the logits. There are some potential questions raised about how well this address the nuanced differences between the $D_{KL}$ objective and the $\| \cdot \|_2$ objective, it is the choice made in previous works (Covert et al., 2023). Alternatives for classification like Shapley-Shubik (Enouen et al., 2024) or Deegan-Packel (Biradar et al., 2024) tend to focus on the one-hot scenario, limiting their application for calibrated prediction. Accordingly, all of our explanations are done on the logits or log-probabilites of the output prediction.

# E  ADDITIVE MODELS

## E.1  PURIFIED LOSS EQUATION

We first reiterate the Fast-Faith-SHAP-k, GAM-k, and Insta-SHAP-GAM-k equations as:

$$\underset{\{\phi_T\}_{T \in \mathcal{I}_{\leq k}}}{\arg\min} \left\{ \mathbb{E}_{x \sim p(x)} \left[ \mathbb{E}_{S \sim p^{\text{SHAP}}(S)} \left[ \left\| f(x; S) - \sum_{T \subseteq [d], |T| \leq k} \mathbb{1}(T \subseteq S) \cdot \phi_T(x) \right\|^2 \right] \right] \right\} \tag{50}$$

$$\underset{\{\phi_T\}_{T \in \mathcal{I}_{\leq k}}}{\arg\min} \left\{ \mathbb{E}_{x \sim p(x)} \left[ \mathbb{E}_{S \sim p^{\text{GAM}}(S)} \left[ \left\| f(x; S) - \sum_{T \subseteq [d], |T| \leq k} \phi_T(x_T) \right\|^2 \right] \right] \right\} \tag{51}$$

$$\underset{\{\phi_T\}_{T \in \mathcal{I}_{\leq k}}}{\arg\min} \left\{ \mathbb{E}_{x \sim p(x)} \left[ \mathbb{E}_{S \sim p^{\text{SHAP}}(S)} \left[ \left\| f(x; S) - \sum_{T \subseteq [d], |T| \leq k} \mathbb{1}(T \subseteq S) \cdot \phi_T(x_T) \right\|^2 \right] \right] \right\} \tag{52}$$

From the Fast-Faith-SHAP-k perspective, the major modification we make is to remove the point-wise flexibility of each of the SHAP-k estimators $\phi_T(x)$ and instead replace each functional approximator with a GAM-like functional approximator $\phi_T(x_T)$. This restricts the capacity of the functional amortizer but as discussed extensively this may give a more accurate representation of the true behavior and also is able to demonstrate improved convergence on synthetic datasets.

From the GAM-k perspective, the major modification is to replace the typical unmasked distribution $p^{\text{GAM}}(S)$ (or sometimes nontrivial in fitting techniques like backfitting), with the masking distribution coming from the Shapley kernel distribution $p^{\text{SHAP}}(S)$. The second key modification we include is the Instant Mask which only allows the additive influence of each function $\phi_T(x_T)$ to flow to the final output so long as all of its constituents have been included in the observed mask $S$. It follows that we may easily calculate downstream explanations of interest like the SHAP value because of the automatic purification of such effects.

## E.2  EXTENSION TO ARBITRARY FRONTIERS

Although the extension of GAMs to higher-order interactions of size 3D and larger is simple to write down as

$$F_{\leq k}(x) \;\; = \sum_{S \in \mathcal{I}_{\leq k}} f_S(x_S), \tag{53}$$

the exploration of these higher-order GAMs delayed because it is typical to be unable to explicitly model all higher-order interaction sets. For instance, the size of $\mathcal{I}_{\leq k}$ grows like $O(d^k)$ which is untenable for most practical purposes. Instead, it has recently been proposed to select only a portion of these interactions as important enough to be included in the model Yang et al. (2020); Dubey et al. (2022); Enouen & Liu (2022). We can consider these more general additive models by first choosing a candidate collection of interactions $\mathcal{I} \subseteq \mathcal{P}([d])$ and then writing the similar equation:

$$F_{\mathcal{I}}(x) = \sum_{S \in \mathcal{I}} f_S(x_S) \tag{54}$$

Once again, we will say the order is the size of the largest subset $k = \max\{|S| : S \in \mathcal{I}\}$; however, there is now a much richer set of choices compared with the original hyperparameter selection of $k$. Hence, despite its simplicity by choosing a small number of feature interactions, it does not provide a reduction in complexity unless we can also answer the question of which feature interactions to include.

## E.3  SOBOL SOLUTION WITH INDEPENDENT VARIABLES

If we briefly return to the case of independent variables, we find that the aforementioned decomposition of variance allows a precise answer to the question of feature interaction selection. Moreover, this means that the functional ANOVA space and the GAM spaces are exactly connected with each other.

$$\underset{\{\phi_T\}_{T \in \mathcal{I}}}{\arg\min} \left\{ \mathbb{E}_{x \sim p(x)} \left[ \left\| F(x) - \sum_{T \in \mathcal{I}} \phi_T(x_T) \right\|^2 \right] \right\} = \sum_{T' \notin \mathcal{I}} \mathbb{V}_{T'} \tag{55}$$

This of course means that if we have a good way to approximate the Sobol indices, then we have an easy way to select for interaction tuples by choosing the largest Sobol indices.

In the case of correlated input variables, we are not so lucky. Although the Sobol covariances (Rabitz, 2010; Hart & Gremaud, 2018) are still able to give a decomposition of the variance of a function

$$\mathbb{V} = \sum_{S \subseteq [d]} \mathbb{C}_S \tag{56}$$

where again $\mathbb{C}_S := \mathbb{Cov}_X[F(X), \tilde{f}_S(X_S)]$, they no longer provide an answer to the effectiveness of an additive model with an arbitrary collection of feature interactions $\mathcal{I} \subseteq \mathcal{P}([d])$. In particular, $\mathbb{C}_S$ may indeed be negative whereas adding an interaction to an additive model can never decrease its represntational capacity. Intuitively, this corresponds to the case where the 'constructive' information provided by allowing a feature interaction is overshadowed by the 'destructive' information created by the redundancies of a feature correlation.

It follows that we must be able to measure the efficacy of an additive model to represent a function under some distribution in an alternative way. In the sections which follow, we will take a variational perspective to represent the efficacy of the interaction collection for an additive model and to begin to answer the question of how to distinguish the multiple types of feature interactions.

### E.4 ADDITIVE MODEL SOLUTION FOR ARBITRARY FRONTIERS

Because of the need to further granulate to the level of synergistic interactions and dissonant interactions, we find that it is necessary to study the entire set of possibilities for additive models. We find that only then can one distinguish between the synergistic interactions and redundant interactions of Figure 2 For $d = 3$, we list all representatives ('frontiers') of nontrivial additive models in Figure 10. For example, the function in Figure 2d would be covered by $1, 2 \equiv \{\emptyset, \{1\}, \{2\}\}$ whereas the function from 2a would need to be covered by $12 \equiv \{\emptyset, \{1\}, \{2\}, \{1, 2\}\}$.

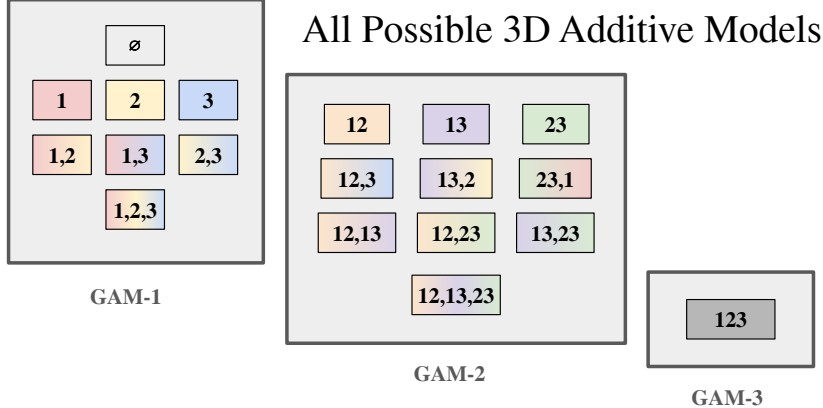

Figure 10: All possible frontiers for a GAM model when $d = 3$.

As mentioned in the main text, one of the critical issues for using additive models to learn a particular target function, is solving the meta-optimization to find an optimal frontier for the additive model. As of yet, there is seemingly no known measurements for the correlated input case paralleling the Sobol indices in the indepdendent input case. In particular, for a given frontier $\mathcal{I} \subseteq \mathcal{P}([d])$ and a candidate interaction $S \subseteq [d]$ not yet in the frontier, there is seemingly no work trying to estimate the differences in errors between these two learned additive models. Moreover, it is noted in the main body that the measurements $C_S$ are in general insufficient to measure these differences in all cases, and must only be used as an approximation.

Herein, we describe the solution to the additive model training procedure as the solution the Euler-Lagrange equation from calculus of variations. Thereafter, we simplify our solution into a single matrix-operator functional equation defined by the projection operators $\mathcal{N}_S$ in the function space

$\mathcal{H}$. We then provide a formal solution to the matrix-operator equation and show how it can be approximated through repeated projections.

**Theorem 6.** Fix an input distribution $X \sim p(X)$ and a function $y = F(x)$. Consider a collection $\mathcal{I} = \{S_1, \ldots, S_L\} \subseteq \mathcal{P}([d])$ and consider the solution to the additive model training equation:

$$\{g_T^*\}_T := \underset{\{g_T\}_T}{\arg\min} \left\{ \mathbb{E}_X \left[ \left\| F(X) - \sum_{T \in \mathcal{I}} g_T(X_T) \right\|^2 \right] \right\} \tag{57}$$

Recalling the conditional expectation projection operators

$$[\mathcal{M}_S \circ F](x) := \mathbb{E}_{\bar{X}_{-S} \sim p(X_{-S}|X_S = x_S)} \left[ F(x_S, \bar{X}_{-S}) \right] \tag{58}$$

where we drop the subscript denoting the distribution $p(x)$.

The solution to the variational GAM training equation in Equation 57 obeys the matrix equation:

$$\begin{pmatrix} e & \mathcal{M}_{S_1} & \cdots & \mathcal{M}_{S_1} \\ \mathcal{M}_{S_2} & e & \cdots & \mathcal{M}_{S_2} \\ \vdots & \vdots & \ddots & \vdots \\ \mathcal{M}_{S_L} & \mathcal{M}_{S_L} & \cdots & e \end{pmatrix} \begin{pmatrix} g_{S_1}^* \\ g_{S_2}^* \\ \vdots \\ g_{S_L}^* \end{pmatrix} = \begin{pmatrix} f_{S_1} \\ f_{S_2} \\ \vdots \\ f_{S_L} \end{pmatrix} \tag{59}$$

*Proof.* Let the objective functional be defined

$$J(\{g_T\}_T) := \mathbb{E}_X \left[ \left\| F(X) - \sum_{T \in \mathcal{I}} g_T(X_T) \right\|^2 \right\} \tag{60}$$

Recall from calculus of variations the Euler-Lagrange equation:

$$\frac{\delta J}{\delta g_{S_i}} \equiv 0$$

for each possible $S_i \in \mathcal{I}$, which then implies:

$$\lim_{\varepsilon_{S_i} \to 0} \frac{1}{\varepsilon_{S_i}} \left[ J(\{g_T + \varepsilon_T \cdot \delta_T\}) - J(\{g_T\}_T) \right] \equiv 0$$

$$\lim_{\varepsilon \to 0} \frac{1}{\varepsilon} \mathbb{E}_X \left[ \left\| F(X) - \sum_{T \in \mathcal{I}} g_T(X_T) - \varepsilon \cdot \delta_{S_i}(X_{S_i}) \right\|^2 - \left\| F(X) - \sum_{T \in \mathcal{I}} g_T(X_T) \right\|^2 \right] \equiv 0$$

$$\lim_{\varepsilon \to 0} \mathbb{E}_X \left[ 2\delta_{S_i}(X_{S_i}) \cdot \left[ F(X) - \sum_{T \in \mathcal{I}} g_T(X_T) \right] + \frac{1}{\varepsilon} \delta_{S_i}^2(X_{S_i}) \right] \equiv 0$$

$$\mathbb{E}_X \left[ 2\delta_{S_i}(X_{S_i}) \cdot \left[ F(X) - \sum_{T \in \mathcal{I}} g_T(X_T) \right] \right] \equiv 0$$

$$\mathbb{E}_{X_{S_i}} \left[ \delta_{S_i}(X_{S_i}) \cdot \mathbb{E}_{X_{-S_i}|X_{S_i}} \left[ F(X) - \sum_{S_i \in \mathcal{I}} g_T(X_T) \right] \right] \equiv 0$$

$$\mathbb{E}_{X_{-S_i}|X_{S_i}} \left[ F(X) - \sum_{T \in \mathcal{I}} g_T(X_T) \right] \equiv 0$$

This can simply be rewritten as:

$$f_{S_i}(X) \equiv \mathcal{M}_{S_i} \circ \sum_{T \in \mathcal{I}} g_T(X)$$

$$f_{S_i}(X) \equiv \mathcal{M}_{S_i} \circ g_{S_i}(X) + \sum_{T \in \mathcal{I} - S_i} \mathcal{M}_{S_i} \circ g_T(X)$$

$$f_{S_i}(x) \equiv g_{S_i}(x) + \sum_{T \in \mathcal{I} - S_i} [S_i \circ g_T](x)$$

$$f_{S_i} = g_{S_i} + \sum_{T \in \mathcal{I} - S_i} [S_i \circ g_T]$$

Hence, it can be seen that each row of the matrix equation corresponds to a partial gradient from the Euler-Lagrange equation as desired. Thus, any possible solution to minimization of the quadratic functional $J$ must obey the above matrix equation. $\qquad\square$

It is moreover the case that we can reduce a collection of feature interactions to its 'frontier' or its solution only on the largest subsets which have no supersets included. From the perspective of the poset $\mathcal{P}([d])$, this corresponds to the set of maximal elements. It can be seen in the above proof that any solution on a projection $S_1 \subseteq S_2$ must automatically be obeyed by the operator equation for the larger set $S_2$.

Accordingly, identify a frontier $\mathcal{I}$ with its set of maximal elements $T_1, \dots, T_{L'}$. Following from Equation 59, we can ensure that it is enough to solve the matrix equation:

$$
\begin{pmatrix} e & \dots & \mathcal{M}_{T_1} \\ \vdots & \ddots & \vdots \\ \mathcal{M}_{T_{L'}} & \dots & e \end{pmatrix} \begin{pmatrix} g^*_{T_1} \\ \vdots \\ g^*_{T_{L'}} \end{pmatrix} = \begin{pmatrix} f_{T_1} \\ \vdots \\ f_{T_{L'}} \end{pmatrix}
\tag{61}
$$

Which we can then take the formal inverse of the operator matrix to yield a solution

$$
\begin{pmatrix} g^*_{T_1} \\ \vdots \\ g^*_{T_{L'}} \end{pmatrix} \text{`` = ''} \begin{pmatrix} e & \dots & \mathcal{M}_{T_1} \\ \vdots & \ddots & \vdots \\ \mathcal{M}_{T_{L'}} & \dots & e \end{pmatrix}^{-1} \begin{pmatrix} f_{T_1} \\ \vdots \\ f_{T_{L'}} \end{pmatrix}
\tag{62}
$$

so long as we take care with the determinant in realizing that a matrix of non-commutative elements does not have a well-defined matrix determinant as it does in the commutative case.

Nonetheless, let us now illustrate the usefulness of such a formal inverse in a simple case with our synthetic example from earlier.

$$
\begin{pmatrix} e & \mathcal{M}_x \\ \mathcal{M}_y & e \end{pmatrix} \begin{pmatrix} g^*_x \\ g^*_y \end{pmatrix} = \begin{pmatrix} f_x \\ f_y \end{pmatrix}
$$

$$
\begin{pmatrix} e & -\mathcal{M}_x \\ -\mathcal{M}_y & e \end{pmatrix} \begin{pmatrix} e & \mathcal{M}_x \\ \mathcal{M}_y & e \end{pmatrix} \begin{pmatrix} g^*_x \\ g^*_y \end{pmatrix} = \begin{pmatrix} e & -\mathcal{M}_x \\ -\mathcal{M}_y & e \end{pmatrix} \begin{pmatrix} f_x \\ f_y \end{pmatrix}
$$

$$
\begin{pmatrix} e - \mathcal{M}_x \mathcal{M}_y & 0 \\ 0 & e - \mathcal{M}_y \mathcal{M}_x \end{pmatrix} \begin{pmatrix} g^*_x \\ g^*_y \end{pmatrix} = \begin{pmatrix} f_x - \mathcal{M}_x \circ f_y \\ f_y - \mathcal{M}_y \circ f_x \end{pmatrix}
$$

$$
\begin{pmatrix} e - \mathcal{M}_x \mathcal{M}_y \\ e - \mathcal{M}_y \mathcal{M}_x \end{pmatrix} \odot \begin{pmatrix} g^*_x \\ g^*_y \end{pmatrix} = \begin{pmatrix} f_x - \mathcal{M}_x \circ f_y \\ f_y - \mathcal{M}_y \circ f_x \end{pmatrix}
$$

$$
\begin{pmatrix} g^*_x \\ g^*_y \end{pmatrix} = \begin{pmatrix} [e - \mathcal{M}_x \mathcal{M}_y]^{-1} \circ [f_x - \mathcal{M}_x \circ f_y] \\ [e - \mathcal{M}_y \mathcal{M}_x]^{-1} \circ [f_y - \mathcal{M}_y \circ f_x] \end{pmatrix}
$$

We can then use the formal Taylor series expansion of the operator inverse to yield:

$$
g^*_x = \sum_{n=0}^{\infty} (\mathcal{M}_x \mathcal{M}_y)^n \circ \left[ f_x - \mathcal{M}_x \circ f_y \right]
$$

$$
g^*_y = \sum_{n=0}^{\infty} (\mathcal{M}_y \mathcal{M}_x)^n \circ \left[ f_y - \mathcal{M}_y \circ f_x \right]
$$

If we choose to denote repeated projections with semicolons, we can then write our solutions as

$$
g^*_x = f_x - f_{y;x} + f_{x;y;x} - f_{y;x;y;x} + f_{x;y;x;y;x} - f_{y;x;y;x;y;x} + \cdots
$$
$$
g^*_y = f_y - f_{x;y} + f_{y;x;y} - f_{x;y;x;y} + f_{y;x;y;x;y} - f_{x;y;x;y;x;y} + \cdots
$$

So then we can caclulate this to be

$$g_x^* = (x + \rho x^2 - \rho) - (\rho^2 x + \rho^3 x^2 + \rho(1 - \rho^2) - \rho) + (\rho^2 x + \rho^5 x^2 + \rho^3(1 - \rho^2) + \rho(1 - \rho^2) - \rho) - \ldots$$
$$= [x] + [-\rho + \rho^3 - \rho^5 + \ldots] + [\rho - \rho^3 + \rho^5 - \ldots]x^2$$
$$= x + \frac{\rho}{1 + \rho^2}[x^2 - 1]$$
$$g_y^* = (\rho y + \rho y^2 - \rho) - (\rho y + \rho^3 y^2 + \rho(1 - \rho^2) - \rho) + (\rho^3 y + \rho^5 y^2 + \rho^3(1 - \rho^2) + \rho(1 - \rho^2) - \rho) - \ldots$$
$$= 0 + [-\rho + \rho^3 - \rho^5 + \ldots] + [\rho - \rho^3 + \rho^5 - \ldots]y^2$$
$$= 0 + \frac{\rho}{1 + \rho^2}[y^2 - 1]$$

It may be checked that this solution agrees with that of directly solving Equation 59:

$$g_x^* = x + \frac{\rho}{1 + \rho^2}[x^2 - 1] \quad g_y^* = \frac{\rho}{1 + \rho^2}[y^2 - 1]$$

It should at the very least be cautioned that these operator manipulations, especially that of the inverse are done only in the formal sense. For instance, considerations of the limit point $\rho = 1$ are not able to demonstrate local convergence in the inversion; however, the formula still remains true in this case. It is considered very likely that these matrix equations are, in most cases, easily able to be solved by the suggested formal manipulations but at least some caution should be exercised.

### E.5 IMPLICATIONS

We reiterate how the additive model's ability to distinguish between synergistic feature interactions and redundant feature interactions is a key strength which has yet to be fully utilized in either the literature on SHAP or the literature on GAMs. By the introduction to the characterization of the set of GAM solutions and drawing parallels with where this aligns and misaligns with the ever-popular functional ANOVA decomposition, we provide a further set of tools to explore SHAP which goes beyond the 'feature-only' perspective of functional ANOVA alone, and begins to explore the 'feature interaction' perspective which adequately handles the intimate complexities which are introduced in the case of correlated variables.

This spectrum of additive models which operates over the entire combinatorially large set of frontiers of additive models is able to give a much more nuanced picture of the underlying structure of both the underlying statistical manifold of the input $X$ variables, but in conjunction with the mapping to the output $Y$ variables. Compared with existing theory in functional ANOVA which spans the entirety of the exponential feature interaction space, this variational formulation covers the range of structures living in the doubly exponential space of all frontiers. As demonstrated in this work, such structure can be directly accessed with relatively simple machine learning approaches, that is additive models and feature masking. It is envisioned that there yet remains many directions of further theoretical exploration to more succinctly and understandably represent the underlying structures of a statistical mapping between data, while simultaneously there still exists abundant opportunities in the application of these learnings directly to machine learning, particularly in bridging the gaps from supervised learning to semi-supervised learning and semi-supervised learning to unsupervised learning.

# F    ADDITIONAL RESULTS

## F.1    ADDITIONAL SYNTHETIC RESULTS

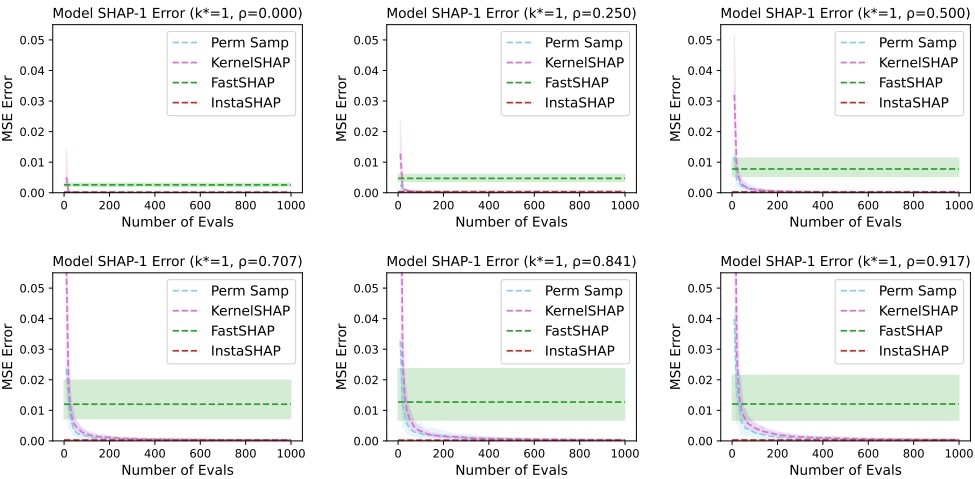

Figure 11: Model MSE Error of SHAP values. Comparison with test-time permutation sampling. $k^* = 1$.

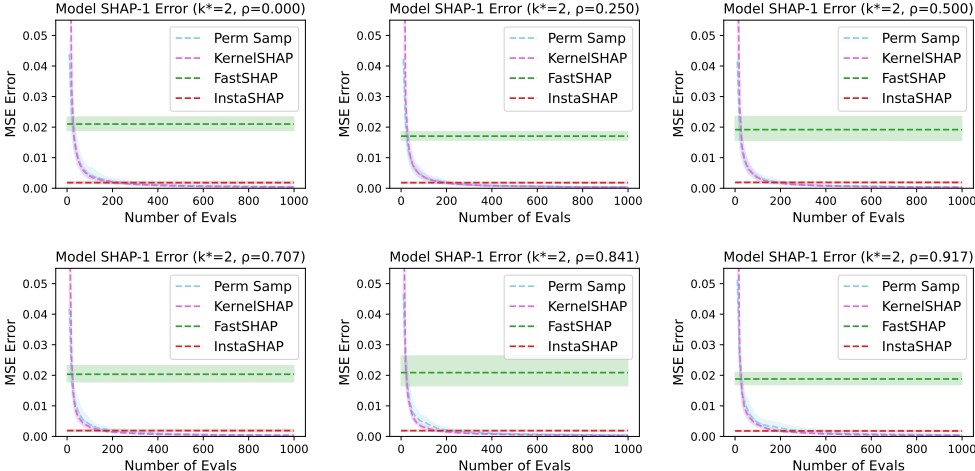

Figure 12: Model MSE Error of SHAP values. Comparison with test-time permutation sampling. $k^* = 2$.

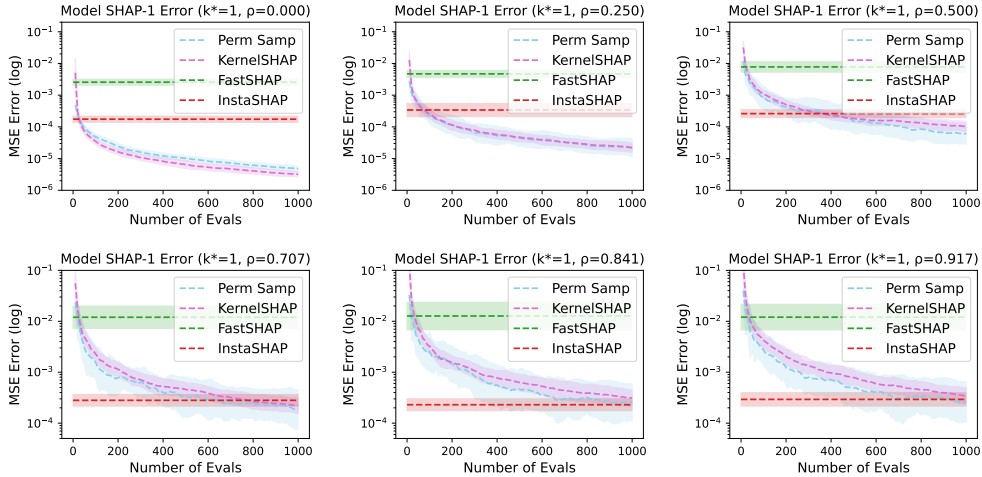

Figure 13: Model MSE Error of SHAP values (logarithmic scale). Comparison with test-time permutation sampling. $k^* = 1$.

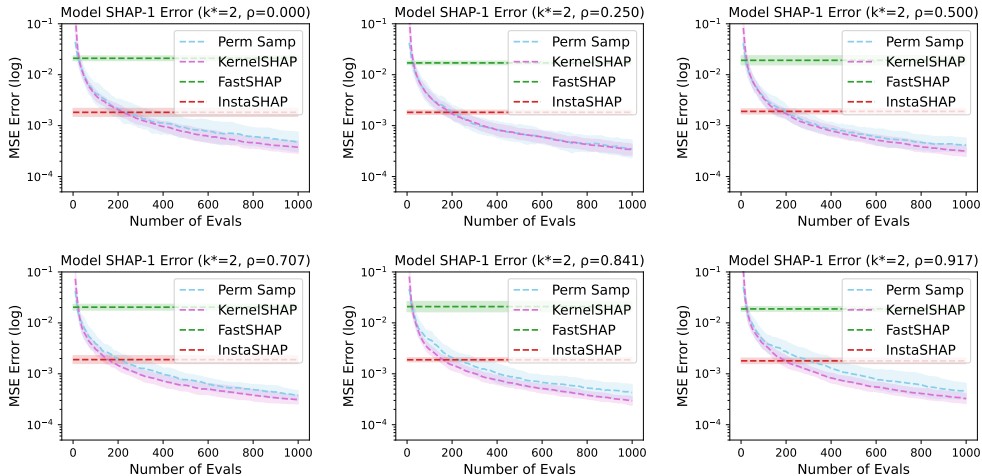

Figure 14: Model MSE Error of SHAP values (logarithmic scale). Comparison with test-time permutation sampling. $k^* = 2$.

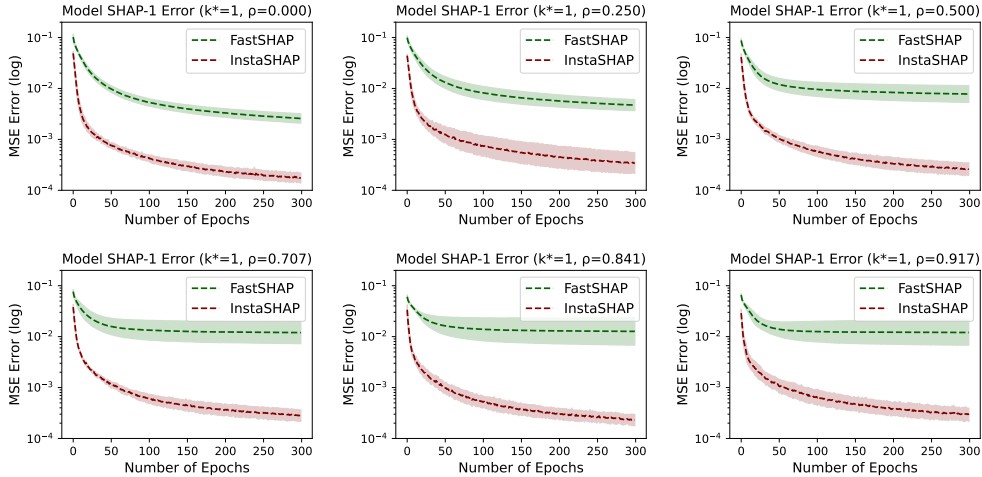

Figure 15: Model MSE Error of SHAP values (logarithmic scale). Comparison with pre-test-time functional amortization. $k^* = 1$.

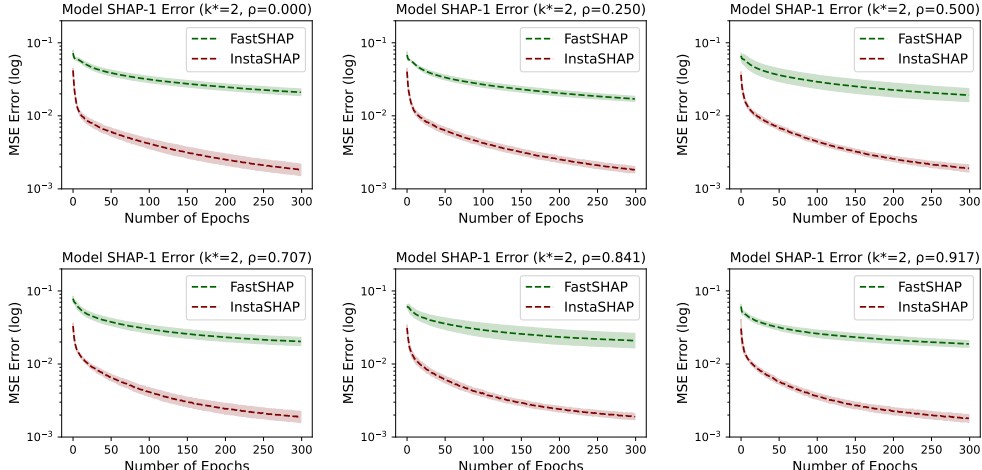

Figure 16: Model MSE Error of SHAP values (logarithmic scale). Comparison with pre-test-time functional amortization. $k^* = 2$.

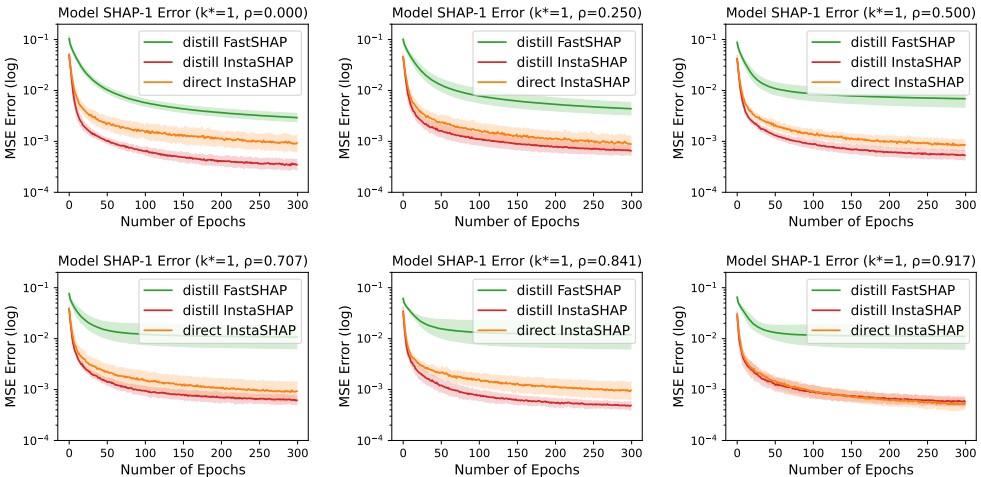

Figure 17: True MSE Error of SHAP values (logarithmic scale). Comparison with pre-test-time functional amortization. $k^* = 1$.

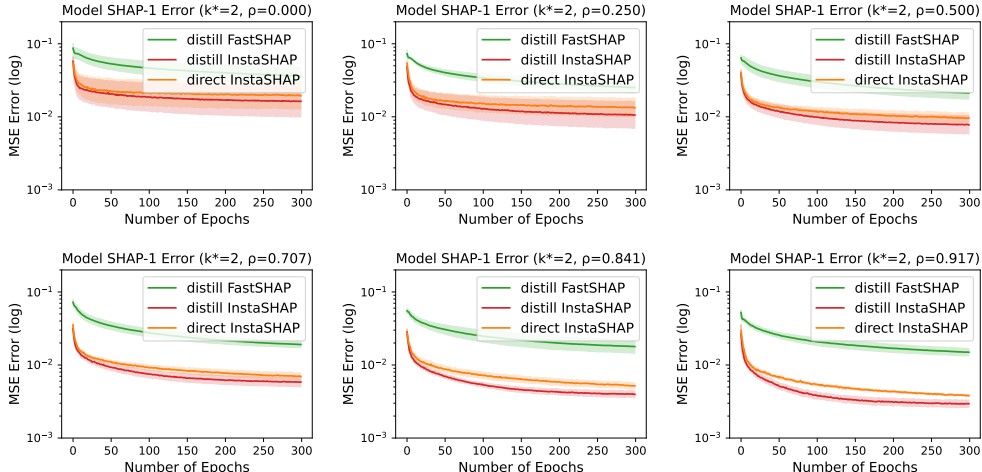

Figure 18: True MSE Error of SHAP values (logarithmic scale). Comparison with pre-test-time functional amortization. $k^* = 2$.

## F.2 ADDITIONAL VISION RESULTS

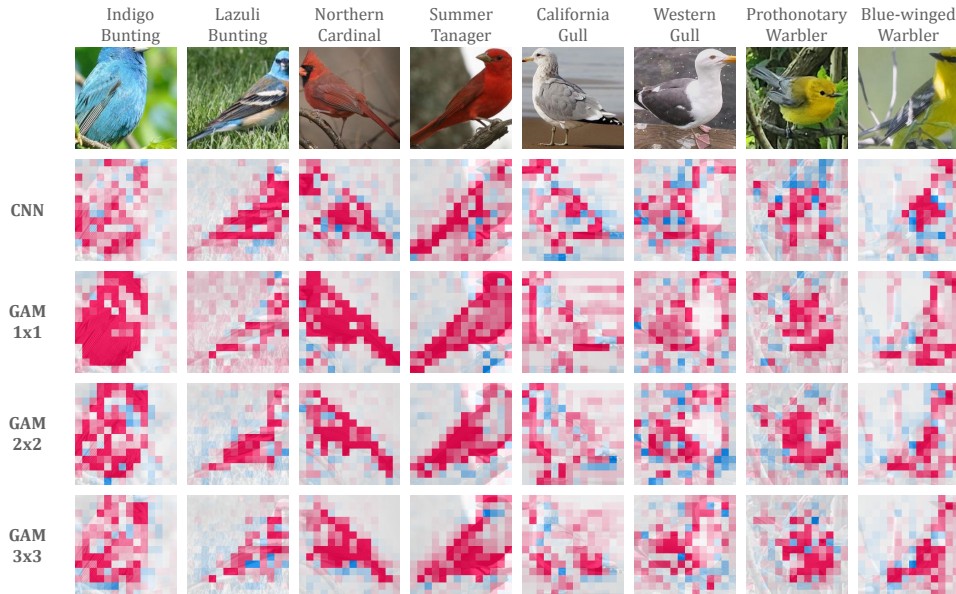

Figure 19: A repeat of Figure 6. Provided for closer comparison with Figure 20 below.

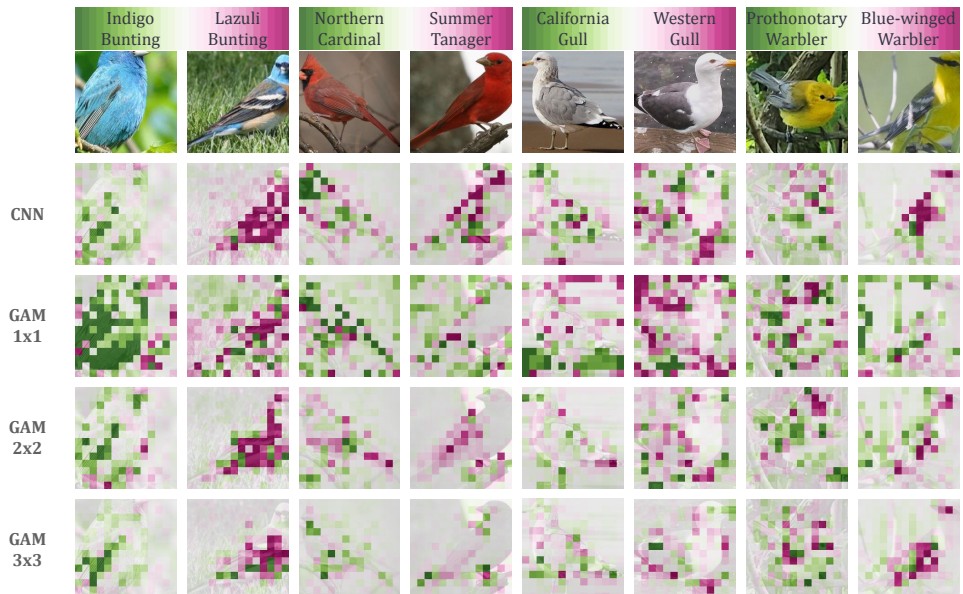

Figure 20: An alternate version of Figure 6 which includes the contrastive Shapley value. In each panel, two similar-looking species are directly compared against one another. For instance, the first two columns compare the Indigo Bunting species (more green) versus the Lazuli Bunting species (more magenta).

In Figure 20, we can see some more granular details about the model explanations with respect to certain species. For instance, the bluer and browner feathers of the two Bunting species, especially in the GAM-1x1 model. Some other characteristics which seem to be picked up by some of the models are the orange beak and black mask of the Cardinal vs. the ordinary beak and face of the Summer Tanager; the yellow feet of the California Gull vs. the orange feet of the Western Gull; and the different upper backs and eye areas for the Prothonotary and Blue-winged Warblers.

### F.3 ADDITIONAL HEALTHCARE RESULTS

We additionally train on a tabular version of the MIMIC healthcare dataset which consists of thirty features used to predict hospital outcomes. We train an ensemble of five additive models using the vanilla GAM training procedure and the InstaSHAP masked training procedure. In Figure 21 below, we display all shape functions learned by the 1D additive model. We plot the mean and one standard deviation according to the ensemble of five models.

The vanilla GAM models achieve accuracies of $91.0\%$, $91.5\%$, $90.6\%$, $91.1\%$, and $91.2\%$ for an average accuracy of $91.1\%$. The InstaSHAP GAM models achieve accuracies of $91.5\%$, $91.3\%$, $91.2\%$, $91.3\%$, $91.0\%$ for an average accuracy of $91.3\%$. Generally, the InstaSHAP models have a more consistent interpretation of the dataset and achieve tighter confidence intervals than the typical training procedure. It can then be assumed a significant amount of the variance between the vanilla ensemble is due to overinterpretaion or sensitivity to the natural correlations of the dataset.

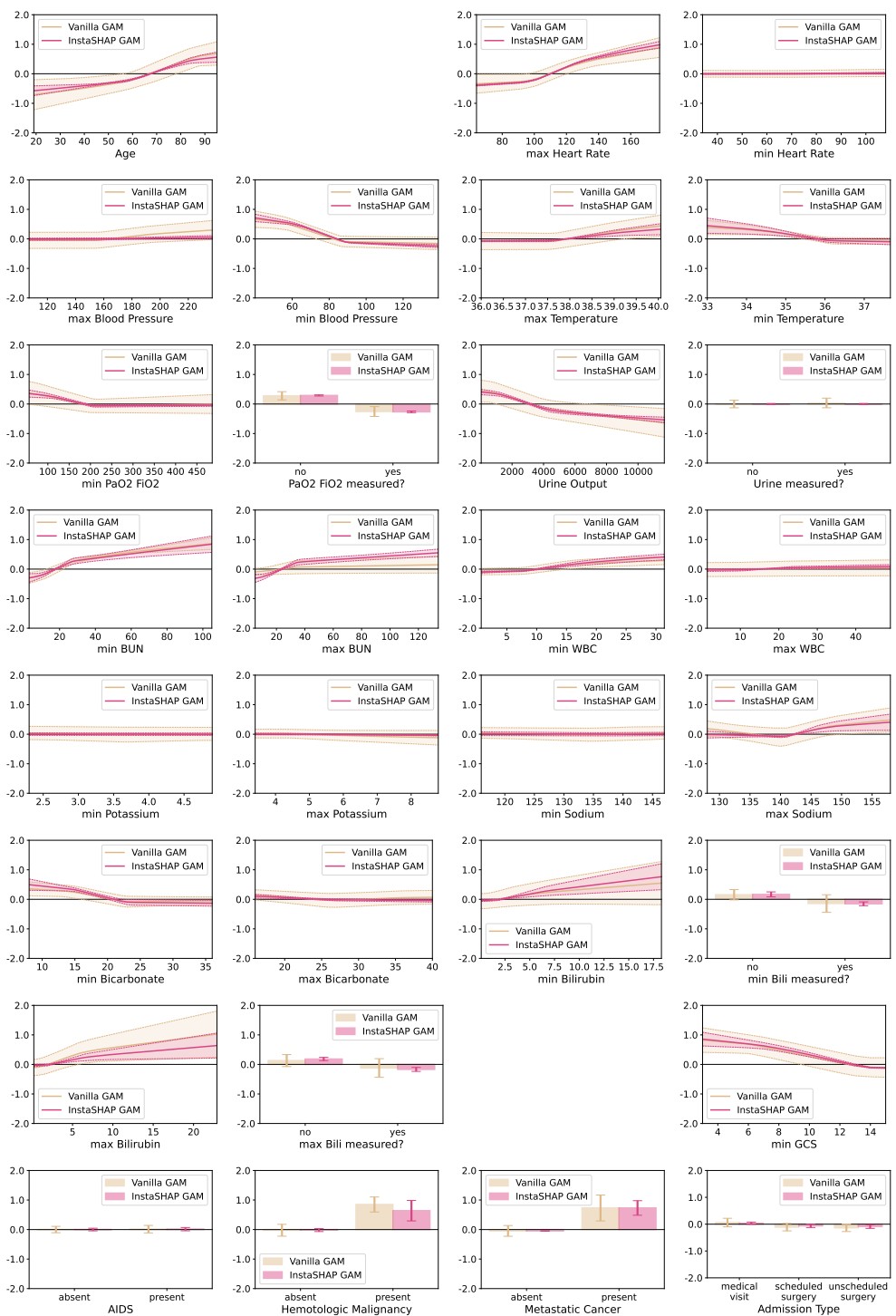

Figure 21: Shape Functions for the MIMIC dataset

## F.4 Additional Finance Results

We train on the census income dataset which consists of thirteen features used to predict whether or not a person's income surpasses a certain level ($50,000 annually). We train an ensemble of five additive models using the vanilla GAM training procedure and the InstaSHAP masked training procedure. In Figure 22 below, we display all shape functions learned by the 1D additive model. We plot the mean and one standard deviation according to the ensemble of five models.

The vanilla GAM models achieve accuracies of $82.1\%$, $84.6\%$, $85.1\%$, $84.4\%$, $84.7\%$, for an average accuracy of $84.2\%$. The InstaSHAP GAM models achieve accuracies of $82.1\%$, $85.4\%$, $84.4\%$, $84.7\%$, $84.7\%$, for an average accuracy of $84.3\%$. We again find that the InstaSHAP models have a more consistent interpretation of the dataset via tighter confidence intervals over the ensemble. Once again, it is assumed that the variance in typically trained GAMs is coming from the inability to consistently interpret the correlations which exist in the dataset.

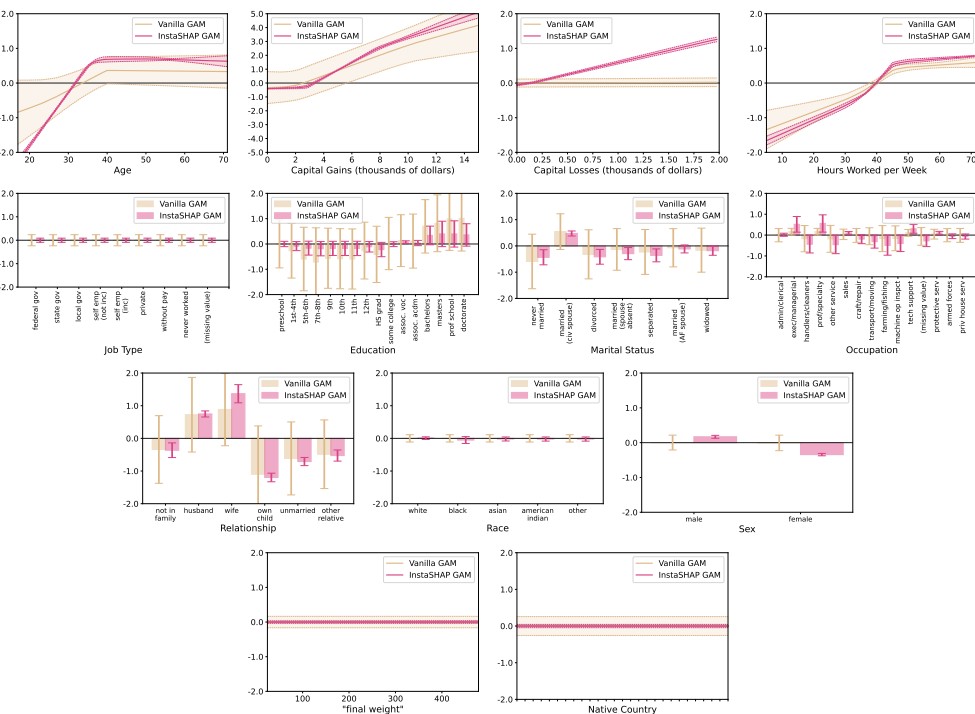

Figure 22: Shape Functions for the Income dataset

