# OpenReview forum: "InstaSHAP: Interpretable Additive Models Explain Shapley Values Instantly"
_ICLR.cc/2025/Conference — ICLR 2025 Poster_

### Official Review · Reviewer_qEEw · 2024-11-03

**Soundness:** 3
**Presentation:** 3
**Contribution:** 2
**Rating:** 6
**Confidence:** 5

**Summary:**

The paper titled "InstaSHAP: Interpretable Additive Models Explain Shapley Values Instantly" presents a novel approach for computing Shapley values efficiently, aimed at improving both the efficiency and interpretability of machine learning model explanations. Though Traditional SHAP methods are powerful, they are often struggle with computational overhead and limited representation capabilities, especially in the presence of feature interactions and correlated inputs.

The authors address these limitations by establishing a theoretical connection between Shapley values and Generalized Additive Models (GAMs), which form the basis for InstaSHAP. InstaSHAP enables the instant calculation of Shapley values in a single forward pass which addresses key computational bottlenecks and making it highly practical for real-time interpretability tasks. The paper’s theoretical contributions unify recent advancements in Shapley-based explanation methods (e.g., FastSHAP, FaithSHAP) through a variational framework that highlights the power of additive models for feature interaction.

The primary contributions of the paper includes, developing a variational formulation that aligns Shapley values with GAMs, providing a new perspective on feature interaction and introducing InstaSHAP, a method that achieves efficient Shapley value computation by purifying GAM models, delivering explanations with significantly improved efficiency and accuracy.

Through extensive experiments on both synthetic and real-world datasets, the paper demonstrates InstaSHAP’s advantages in capturing complex feature interactions and delivering reliable, interpretable explanations at scale. This work contributes to advancing interpretable machine learning by addressing the practical constraints of existing SHAP-based methods, which makes the way for broader application of explainable AI in high-dimensional and high-stakes environments.

**Strengths:**

Originality: This paper brings a fresh approach by combining Shapley values with Generalized Additive Models (GAMs) to create InstaSHAP, which allows Shapley values to be computed in a single forward pass. This is an impressive alignment of two core ideas in model interpretability—using Shapley values for feature importance and GAMs for creating interpretable models. By introducing a variational framework that connects GAMs with Shapley explanations, the authors present a conceptually rich and practical model for instant interpretability. This work also does a great job addressing and combining limitations of methods like FastSHAP and FaithSHAP, enabling it to tackle feature interactions even in high-dimensional datasets.

Quality: The paper’s quality is high, with robust theoretical analysis backed by thorough empirical validation. The authors prove several key points, including the link between Shapley values and functional ANOVA decompositions, and specify conditions where Shapley explanations hold up. They support these theoretical findings with extensive experiments on synthetic and real-world datasets, showing how InstaSHAP outperforms traditional methods both in speed and accuracy. Overall, the paper is well-researched, thoughtfully designed, and clearly accounts for the limitations and assumptions behind their approach, giving a well-rounded view of InstaSHAP’s potential.

Clarity: The paper is generally well-written, guiding the reader from theoretical underpinnings to the practical use of InstaSHAP. Key ideas, like the interplay between Shapley values and GAMs, are explained in detail, making it easier to understand how the method works even for readers less familiar with the nuances of either technique. Visual aids and detailed breakdowns of the mathematics further aid clarity. However, some heavily mathematical sections could benefit from a more intuitive explanation to make it even more accessible, especially for readers focused on practical applications.

Significance: This work could make a meaningful impact on the field of interpretable machine learning. InstaSHAP’s efficiency makes Shapley-based explanations much more practical for real-time use in areas like healthcare, finance, and other high-stakes domains. Its ability to handle feature interactions through the GAM framework could inspire further development in interpretable models that are both fast and powerful. The theoretical contributions here also lay a solid foundation for future work on GAMs and functional ANOVA in model interpretation, which researchers and practitioners could build on.

**Weaknesses:**

Limited Scope of Experiments

The experiments on synthetic and tabular datasets give a good picture of InstaSHAP’s strengths, but the evaluations on high-dimensional data, like images or text, feel a bit limited. Given that InstaSHAP is designed for complex tasks in fields like computer vision (CV) and natural language processing (NLP), it would be helpful to see its performance on a broader range of real-world datasets in these areas. Right now, the CUB bird dataset is the only high-dimensional example, so including more CV and NLP datasets could provide a stronger case for InstaSHAP’s versatility in real-world applications.

Suggestion: Adding evaluations on popular high-dimensional datasets in CV (like CIFAR-10 or ImageNet) and NLP (such as SST-2 or IMDB) would better showcase InstaSHAP’s performance on tasks that feature complex interactions and correlations.

Complexity of Theoretical Explanations

The paper does an impressive job with the theory, but some sections—especially those that dive into the variational equations and functional ANOVA—might feel overwhelming for readers who aren’t well-versed in the math. It would be helpful to have more intuitive explanations in these areas, so readers can grasp the main ideas without needing to go too deep into the technical details.

Suggestion: Including a high-level summary or simpler examples to illustrate the variational formulation and ANOVA concepts could make these sections more accessible. Even a short summary or diagram could help clarify the main points for readers who are looking for an overall understanding rather than technical details.

Assumptions About Feature Correlations

InstaSHAP’s approach assumes certain patterns in how features are correlated, especially in how it handles feature interactions. While this is touched upon, the paper could go further in explaining how InstaSHAP might perform if the correlations between features differ from these assumptions. This would be valuable for practitioners who want to apply InstaSHAP to datasets with different correlation structures.

Suggestion: Providing a discussion or some analysis of InstaSHAP’s performance on datasets with varied correlation patterns could be helpful. Even some general guidelines on when InstaSHAP might face limitations with unusual or weaker correlations would offer more context for its practical use.

Limited Comparison with Other Explainability Methods

The paper primarily compares InstaSHAP with other SHAP-based methods like FastSHAP and FaithSHAP, but it would be helpful to see how it stacks up against non-SHAP explainability methods like LIME, Integrated Gradients, or counterfactual explanations. This would give a more complete view of InstaSHAP’s strengths and where it might still have limitations.

Suggestion: Adding comparisons with a few well-known, non-SHAP explainability techniques could give a more rounded picture of where InstaSHAP fits into the broader field. Showing InstaSHAP’s performance alongside these other methods on selected benchmarks could highlight its unique advantages or challenges.

Scalability to Higher-Order Interactions

The paper points out that InstaSHAP has trouble scaling with higher-order interactions in high-dimensional data, which impacts its performance on tasks with complex dependencies, as seen in the bird classification experiment. Since handling these kinds of interactions is important for CV tasks with spatial relationships or NLP tasks with sequential data, it would be great if InstaSHAP could be adapted to handle these better.

Suggestion: Exploring modifications to InstaSHAP’s architecture, like using hierarchical or multi-scale structures, could potentially improve its handling of complex dependencies. Even some suggestions for future work in this area would be helpful.

**Questions:**

1. Additional High-Dimensional Datasets in Evaluation

    Question: Could you provide more insights into why the high-dimensional evaluations were limited to the CUB bird dataset? Was it a matter of computational resources, dataset availability, or other considerations?
    Suggestion: Including results on a broader set of high-dimensional datasets, such as CIFAR-10, ImageNet for computer vision, or popular NLP datasets (e.g., SST-2 or IMDB), would help demonstrate InstaSHAP’s scalability and effectiveness across a wider range of real-world scenarios. If adding these evaluations is not feasible, a discussion on InstaSHAP’s potential limitations or expected behavior on these kinds of datasets would be beneficial.

2. Intuitive Explanation of the Theoretical Framework

    Question: Some sections, particularly those with dense variational equations and functional ANOVA decompositions  may be challenging for readers without a strong theoretical background. Could you clarify or add a more intuitive explanation of the variational framework and functional ANOVA, perhaps with a real-world example or simplified illustration?
    Suggestion: A high-level summary or visual aid, either in the main text or supplementary material, could make the theoretical foundations more accessible. This could help readers, especially practitioners, grasp the core concepts without needing to follow every technical detail.

3. Handling of Feature Correlations

    Question: InstaSHAP relies on certain assumptions about feature correlations, particularly with respect to how it handles conditional expectations in feature interactions. Could you elaborate on how InstaSHAP might perform in cases where correlations differ significantly from these assumptions? Are there any specific conditions under which InstaSHAP’s accuracy or interpretability might be compromised?
    Suggestion: A more detailed discussion on how InstaSHAP manages datasets with varying levels of feature correlation would be valuable. It would also help to provide guidelines on how practitioners can assess whether their datasets align with InstaSHAP’s assumptions, or whether InstaSHAP might be less effective under certain correlation structures.

4. Comparison with Non-SHAP Explainability Methods

    Question: The paper mainly compares InstaSHAP with other SHAP-based methods. Could you provide more context on how InstaSHAP would perform in comparison to other interpretability methods, such as LIME, Integrated Gradients, or counterfactual explanations? For example, what are its expected advantages or limitations relative to these non-SHAP approaches?
    Suggestion: Adding experiments or a comparative analysis with non-SHAP methods could offer a fuller understanding of InstaSHAP’s relative performance and strengths. This would also give readers insight into when InstaSHAP might be preferable to these other methods and when it might be more limited.

5. Scalability to Higher-Order Interactions and Complex Dependencies

    Question: The paper mentions challenges in scaling InstaSHAP for datasets with higher-order interactions or complex dependencies, which is especially relevant in tasks with spatial or sequential data. Do you see potential ways InstaSHAP could be adapted to handle these scenarios more effectively?
    Suggestion: Exploring architectural modifications or possible future improvements for handling higher-order interactions, such as multi-scale or hierarchical structures, would be helpful. Even speculative ideas on how InstaSHAP could evolve in this direction could be useful for readers interested in expanding its applicability.

6. Practical Guidelines for Real-World Application

    Question: Are there specific types of datasets or problem settings where InstaSHAP might be particularly well-suited or less effective? Do you have recommendations for practitioners on the best practices for applying InstaSHAP in real-world tasks?
    Suggestion: Including a section with practical guidelines, detailing ideal use cases or specific considerations for InstaSHAP’s application in real-world scenarios, would provide valuable context for practitioners. This might include advice on dataset types, feature characteristics, or model architectures that align well with InstaSHAP’s strengths.

---

> ### Author Response · Authors · 2024-11-17
>
> > Could you provide more insights into why the high-dimensional evaluations were limited to the CUB bird dataset?
>
> >  Adding evaluations on popular high-dimensional datasets in CV (like CIFAR-10 or ImageNet) and NLP (such as SST-2 or IMDB) would better showcase InstaSHAP’s performance…  a discussion on InstaSHAP’s potential limitations or expected behavior on these kinds of datasets would be beneficial.
>
> High-dimensional experiments were limited to the CUB dataset because we feel it should be completely representative of the larger phenomenon we identify which is underpinning a vast swath of the existing SHAP literature, particularly in high-dimensional CV/NLP datasets.  This is in contrast to the GAM literature which has not seen widespread usage in similar high-dimensional tasks.  Some of the closest existing attempts are [3] which do not operate on the input pixel feature in the same way as SHAP, but rather work with CNN extracted features, meaning such GAMs are not relevant for this work.
>
> In terms of comparison with other CV datasets, our methods would easily extend and we would expect behavior to follow the trend of larger and more complex datasets like ImageNet having even more difficulties with applying GAM models, further proving our hypothesis.
>
> In NLP, there is a major barrier to applying GAM methods because of variable length inputs.  The common technique of handling these is through the transformer architecture; however, there do not yet exist attempts to merge a local GAM structure with a variable input architecture like the transformer.  Although we expect the findings would be extremely similar.  For instance, bag of words methods (similar to GAM 1 methods) and bigram methods (similar to GAM 2 methods) have long been out of date and cannot achieve modern accuracies.
>
> > …might feel overwhelming for readers who aren’t well-versed in the math
> without needing to go too deep into the technical details.
>
> > simpler examples to illustrate the variational formulation
>
> The words “variational” we use throughout, for those without a theoretical background, in mathematics is similar in nature to “optimization” in computer science terms.  “variational” is used to emphasize that the optimization is taking place over a functional space rather than the typical Euclidean space ($R^d$).  That is to say KernelSHAP uses an optimization based definition of the Shapley value and FastSHAP uses a variational (functional optimization) based definition of the Shapley value.  These are perhaps obvious to an expert in the related works, such as yourself; however, we have added flavor text to Section 4 to enhance its readability.
>
>
> > InstaSHAP relies on certain assumptions about feature correlations,
>
> The authors **simply cannot** understand this question.  There are absolutely no assumptions about the feature correlations in the dataset.  This is the particular advantage of InstaSHAP when compared to all existing previous works, especially compared with those focusing on theoretical developments.  Moreover, this is the key development when compared to the previous work [1].  Can the reviewer please clarify what assumptions are made about the input feature correlations and/or how other previous works somehow make less assumptions than our work?
>
> > how InstaSHAP would perform in comparison to other interpretability methods, such as LIME, Integrated Gradients, or counterfactual explanations?
>
> The authors do not believe that additional comparison with LIME, IG, or counterfactual explanations would provide significant value to the work.  In particular, when the reviewer says they are interested in the ``performance’’ of each method, can they elaborate on what is meant by this?
>
> > The paper points out that InstaSHAP has trouble scaling with higher-order interactions in high-dimensional data, which impacts its performance on tasks with complex dependencies, as seen in the bird classification experiment.
>
> > Do you see potential ways InstaSHAP could be adapted to handle these scenarios more effectively?
>
> Yes, InstaSHAP is a GAM method.  GAMs previously have never had success in these complex tasks like CV and NLP.  This limitation is the key point we highlight in this work.  By highlighting this fact, we provide an ultimatum: use something other than SHAP or adapt GAMs to handle these complexities.  We see such potential ways to handle these scenarios as future work for the GAM literature enabled by our theoretical insights.
>
> > Including a section with practical guidelines, detailing ideal use cases or specific considerations for InstaSHAP’s application in real-world scenarios, would provide valuable context for practitioners.
>
> We have added an additional subsection to the end of our theory section, Section 3, which focuses on the practical interpretation of the results as is highlighted in Figure 1.
>
> [3] “Scalable Interpretability via Polynomials” Abhimanyu Dubey et al. 2022.

---

> > ### Comment · Reviewer_qEEw · 2024-11-25
> >
> > Thanks for your responses. The questions that I have asked has been fairly clarified with your responses.Regarding, more evaluation, it has been clarified with results from Appendix.Although my doubts regarding the scalability and the real world applications remains a  little vague. Overall the work is fair and the experimental results shows it.

---

> > > ### Author Response · Authors · 2024-11-26
> > >
> > > We are happy to have been able to address most of the concerns written in your name.  It would be friendly of you to lower your confidence to reflect reality.

---

> > > > ### Author Response · Authors · 2024-11-28
> > > >
> > > > For your reference, usually if the reviewer is not able to find time to read the paper, they would be inclined to put a confidence of 1 or 2.

---

### Official Review · Reviewer_kSd9 · 2024-11-03

**Soundness:** 2
**Presentation:** 3
**Contribution:** 2
**Rating:** 6
**Confidence:** 5

**Summary:**

The paper "InstaSHAP: Interpretable Additive Models Explain Shapley Values Instantly" introduces InstaSHAP, an efficient method for calculating Shapley values to explain machine learning model predictions. InstaSHAP leverages a novel connection between Shapley values and Generalized Additive Models (GAMs), allowing Shapley values to be computed in a single forward pass. This variational approach addresses computational bottlenecks in traditional SHAP methods and improves accuracy in scenarios with complex feature interactions.

Key contributions include a unified framework that connects GAMs and Shapley explanations, the development of InstaSHAP for fast Shapley computation, and extensive experiments showing its practical advantages over other methods like FastSHAP. This work makes Shapley-based explanations more accessible for real-time applications in high-stakes areas such as healthcare and finance which advancing the field of interpretable machine learning.

**Strengths:**

Originality: The paper uniquely links Shapley values and Generalized Additive Models (GAMs), presenting an innovative approach to model interpretability. Introduction of "InstaSHAP" enables real-time Shapley value computation through GAM-based training which stands out as an original contribution.

Quality: Strong, rigorous proofs link SHAP and GAMs can provide a detailed understanding of representation limits and power. Validated on synthetic, tabular, and image datasets showcase clear improvements over existing methods like FastSHAP and FaithSHAP.

Clarity: In this paper, complex concepts are made accessible with formal mathematical definitions and visual aids. Comprehensive context on Shapley values and GAMs ensures readers understand the motivation and methods.

Significance: The paper addresses gaps in traditional SHAP methods, especially for correlated and high-dimensional data, with real-time applicability. InstaSHAP’s instant computation is highly relevant for fields needing quick and accurate model insights, like healthcare and finance.

**Weaknesses:**

Empirical Analysis: More diverse datasets (e.g., time series, medical) would better demonstrate robustness. A broader comparison with methods like Integrated Gradients or LIME is needed.

Scalability: The paper lacks details on computational efficiency for large datasets. Quantitative analysis of runtime and latency is needed to validate real-time claims.

Theoretical Depth:  More discussion on handling complex non-linear dependencies would clarify limitations. The method's performance in non-GAM-friendly tasks needs clearer boundaries.

Clarity: Intuitive explanations alongside proofs would improve accessibility. More detailed captions and annotations are needed for clarity.

**Questions:**

Questions:
Validation: Would you include experiments with more varied data types (e.g., time series, medical)? Have you considered comparing with Integrated Gradients or LIME?

Scalability and Performance: How does InstaSHAP scale on large datasets compared to FastSHAP? Can you share latency and runtime benchmarks?

Handling Complexity: How does InstaSHAP manage complex, high-order dependencies? Have you tested it in scenarios where GAMs typically underperform?

Clarity Enhancements: Would you add intuitive explanations alongside complex proofs? Could you improve figure captions for clarity? Any real-world cases showing InstaSHAP's advantages? Are there plans for easy implementation or integration with SHAP tools?

Suggestions

Expand Empirical Analysis: Include experiments on more varied datasets to showcase broader applicability and compare InstaSHAP with methods like Integrated Gradients or LIME for a comprehensive evaluation.

Provide Scalability Insights: Add benchmarks on training times and real-time performance metrics to support the paper’s scalability and real-time claims.

Clarify Theoretical Scope: Discuss how InstaSHAP handles high-order, non-linear interactions and elaborate on its limitations with complex models.

Enhance Clarity: Add intuitive explanations for theoretical sections and improve figure annotations to make the paper more accessible.

Highlight Practical Use: Present real-world case studies demonstrating InstaSHAP's benefits and share plans for making it user-friendly (e.g., open-source tools).

---

> ### Author Response · Authors · 2024-11-17
>
> > More diverse datasets (e.g., time series, medical) would better demonstrate robustness.
>
> We provide additional experiments to the end of the appendix demonstrating our insights are consistent across more diverse tabular datasets including those coming from the medical and financial domains.
>
>
> > A broader comparison with methods like Integrated Gradients or LIME is needed.
>
> Can the reviewer please explain in their own words why they believe a comparison with Integrated Gradients or LIME is necessary?
>
> > The paper lacks details on computational efficiency for large datasets. Quantitative analysis of runtime and latency is needed to validate real-time claims.
>
> First, to be clear, no runtime performance is necessary to `validate’ the real-time claims.  Real-time claims are in accordance with the paradigm introduced by FastSHAP and furthered by our work InstaSHAP.  In that sense, real-time is in reference to the lack of test-time sampling required by previous SHAP approaches instead of the functional amortization approach.  Nevertheless, benchmarks are provided for your convenience: a Resnet50 CNN takes 6.53 hours to finetune on CUB for 300 epochs; a InstaSHAP Resnet18 GAM takes 6.58 hours to finetune on CUB for 300 epochs, both on a 2080 Ti GPU.
>
> > The method's performance in non-GAM-friendly tasks needs clearer boundaries.
>
> > How does InstaSHAP manage complex, high-order dependencies?
>
> > Have you tested it in scenarios where GAMs typically underperform?
>
> Yes, we have tested our method in scenarios where GAMs typically underperform.  GAMs have never had success in high-dimensional CV tasks.  CUB is a high-dimensional CV task.  It is one of our main thesis points that this is a previously unidentified problem existing in the literature.  If this is not sufficient explanation, can you please explain what is meant by you in this question?
>
>
> > Intuitive explanations alongside proofs would improve accessibility.
>
> We have added a third subsection to our theory section which gives a brief summary of the results to be more easily interpreted by a practitioner.
>
> > Could you improve figure captions for clarity?
>
> We have improved the captions of Figure 1 and the figure itself slightly.  Hopefully, the caption will read more clearly.  More specific feedback on what is meant by this suggestion would also be useful.
>
> > Are there plans for easy implementation or integration with SHAP tools?
>
> Yes, our method will be made open source and integrated with common libraries to allow for ease of use.

---

> > ### Comment · Reviewer_kSd9 · 2024-11-26
> >
> > Thanks for your detailed clarifications. It is very appreciated. However, It is not that much clear to me about how does InstaSHAP can manage complex, high-order dependencies.

---

> > > ### Author Response · Authors · 2024-11-26
> > >
> > > You claim to have perfect familiarity with all of the related works (GAM, SHAP, feature interactions, functional ANOVA); however, somehow you are not able to form this question as if you have read the paper.  In order to nevertheless enable discussion on the topic, we will list a series of facts, and you can choose which one is unclear or causing confusion:
> > >
> > > - CV is a task with complex, high-order dependencies
> > >
> > > - GAMs have never had success in high-dimensional CV tasks
> > >
> > > - higher order interactions require higher order GAMs
> > >
> > > - we test our method in domains where GAMs succeed (tabular)
> > >
> > > - we test our method in domains where GAMs fail (CV)
> > >
> > > - our InstaSHAP method applies to all GAMs whether they succeed or fail in achieving SOTA performance
> > >
> > > - GAM SHAP values disagree with CNN SHAP values on complex, high-order tasks

---

### Official Review · Reviewer_pfXM · 2024-11-05

**Soundness:** 4
**Presentation:** 3
**Contribution:** 3
**Rating:** 8
**Confidence:** 4

**Summary:**

The paper introduces InstaSHAP, a method that leverages the link between Shapley values and Generalized Additive Models (GAMs) to create a more efficient model for interpreting ML models. The authors argue that while SHAP explanations are widely used for model interpretability, they have inherent computational challenges and limitations in representing feature interactions. The authors propose a variational approach that unifies Shapley values with additive models via functional ANOVA. InstaSHAP provides an approach to explain models in a single forward pass, and incorporates a new loss function with output masking to automatically purify GAM models. Experiments demonstrate that InstaSHAP outperforms baselines in approximating Shapley values on synthetic, tabular, and high-dimensional data.

**Strengths:**

This paper introduces InstaSHAP, an approach that unifies Shapley value explanations with Generalized Additive Models (GAMs) to improve computational efficiency and representation for model interpretation. This work is original in its use of functional ANOVA to connect Shapley values and GAMs, and in addressing limitations in handling correlated features—an area where Shapley methods traditionally struggle.

The paper is technically sound, presenting a useful theoretical framework with a loss function for automatic GAM purification. The experiments are well-designed, testing InstaSHAP across synthetic, tabular, and high-dimensional datasets, with results (at least on synthetic data) that convincingly support the method’s advantages over existing approaches like FastSHAP in representing complex feature interactions.

**Weaknesses:**

1. The experiments could benefit from more varied, real-world datasets that reflect the complexity of correlated feature spaces in practical applications. For example, exploring datasets in domains like finance or healthcare—where feature correlations are known to impact interpretability—could showcase the utility of InstaSHAP in settings beyond the selected synthetic and high-dimensional benchmarks. Additionally, while the authors compare InstaSHAP to FastSHAP, further comparisons with other recent Shapley-based methods, such as KernelSHAP or TreeSHAP, could provide a more comprehensive evaluation of the method’s relative strengths and weaknesses.

2. While the InstaSHAP method addresses the efficiency limitations of SHAP, it would be helpful to clarify any potential trade-offs in accuracy or interpretability when using the purified GAM approach. Addressing when and why InstaSHAP might yield different Shapley values from those generated by traditional methods could enhance readers' understanding of the practical implications and limitations of this approach.

3. The paper could benefit from a discussion on how InstaSHAP’s approach relates to causal explanation methods. For instance, the authors could consider highlighting situations where InstaSHAP may not capture causal relationships due to feature correlations that stem from confounding variables rather than true causal links.

**Questions:**

1. Currently, InstaSHAP is mainly compared with FastSHAP. Would you consider adding comparisons with other popular Shapley-based methods like KernelSHAP or TreeSHAP?

2. Given that InstaSHAP uses purified GAM models to approximate Shapley values, are there cases where this might result in different Shapley attributions compared to traditional methods?

3. Shapley-based methods, including InstaSHAP, are associational rather than causal. How do you see this gap affecting the performance of InstaSHAP? How can you deal with sensitivity to at least observed confounders?

3.  The experiments are useful, yet most real-world evaluations focus on high-dimensional image data. Did you consider expanding InstaSHAP’s evaluation to real-world datasets where correlated feature spaces play a more critical role?

---

> ### Author Response · Authors · 2024-11-17
> **Initial Author Response (part 1/2)**
>
> > Would you consider KernelSHAP or TreeSHAP?
>
> We cannot easily consider TreeSHAP because it is an architecture-specific SHAP approach, and our target high-dimensional applications (e.g. CUB) cannot be solved easily by tree-based machine learning approaches.  From a modern lens, however, TreeSHAP could be considered a special case of [1] and [1] could be considered a special case of our work (where the key insights is that for trees, the max depth automatically restricts the maximum size of feature interactions, $k$ in our work).
>
> We can consider KernelSHAP and add it as another baseline like permutation sampling, although generally this approach has slightly fallen out of favor because its two stage procedure is more difficult to use than permutation sampling.  These results are updated in Appendix F.  Generally, we see that the performance tracks very closely to the permutation sampling approach, although appearing to be slightly more stable on this synthetic data.
>
>
> > Given that InstaSHAP uses purified GAM models to approximate Shapley values, are there cases where this might result in different Shapley attributions compared to traditional methods?
>
> It is maybe first important to remember that there is only one Shapley value after fixing the model to explain; however, there are many various ways to approximate it.  For the synthetic datasets where we benchmark different approximations, the methods we compare are almost always getting errors less than 1.0e-2.  In that context, although our method is better and/ or faster at calculating the Shapley value, there is not a huge difference in the final attributions compared to traditional methods.
>
> On the other hand, there **is a large gap** between the Shapley values of a blackbox model and the Shapley values of a GAM model.  We specifically identify this on the high-dimensional CUB dataset.  In Figure 6, you can see the Shapley values for a CNN (calculated with permutation sampling to ensure no GAM bias is induced) vs. the Shapley values for three different GAM models (calculated with InstaSHAP) have a significant gap in their explanations.  This difference we theoretically identify as a fundamental limitation of Shapley, meaning that there must be information which is being destroyed by the Shapley value.  Moreover, from a practitioner’s standpoint, seeing that the CNN and GAM-1x1 have different Shapley explanations on this dataset implies that the practitioner cannot completely trust the CNN Shapley values.

---

> > ### Author Response · Authors · 2024-11-17
> > **Initial Author Response (part 2/2)**
> >
> > > Shapley-based methods, including InstaSHAP, are associational rather than causal. How do you see this gap affecting the performance of InstaSHAP? How can you deal with sensitivity to at least observed confounders?
> >
> > First, the SHAP based approaches we consider (baseline, marginal, observational) will never find causal relationships.  These methods only explain the associational relationships learned by the blackbox model.  We, the authors, strongly believe that causal insights require causal assumptions.  The most popular of these assumptions for the SHAP literature is the assumption that every input feature is completely independent from one another [2].  We feel this is an extremely restrictive assumption which precludes finding causal insights on basically every real-world machine learning dataset.  Simultaneously, the authors envision this work as one of the first serious theoretical attempts to push beyond the restrictiveness of the feature independence assumption for SHAP compared with existing works [1,2].
> >
> > In that sense, although our work does not directly solve the problem of causal explainability, in the case of observed confounders you mention, our work does make significant steps down this path.  In particular, observed confounders from the causality literature are the exact same as feature correlations from the explainability literature.  Accordingly, our method which handles “redundant interactions” has many parallels with dealing with “observed confounding” so long as one is operating under the ‘no hidden confounding’ assumption.  Although there still remain many interesting questions to be answered from a causal perspective, perhaps this work helps to highlight the connection and opens many future opportunities in this direction.
> >
> > Perhaps if you also have some more specific concerns or interests in mind, we could more directly compare and contrast them against InstaSHAP.
> >
> >
> >
> > > The experiments are useful, yet most real-world evaluations focus on high-dimensional image data. Did you consider expanding InstaSHAP’s evaluation to real-world datasets where correlated feature spaces play a more critical role?
> >
> > It is not at first clear to the authors what you mean.  We believe feature correlations play a critical role in our understanding of image data.  In fact, it is commonly touted that image data is so correlated that it likely spans a strictly lower dimensional subspace of the space of all possible input images (image manifold hypothesis). This does have a significant impact on the interpretation of SHAP for image data which has not been thoroughly explored before.
> >
> > Nevertheless, within the context of the previous questions about healthcare/finance datasets and also causal insights, it is assumed that there is an interest in understanding datasets where feature correlations play a more **direct role** in the interpretation like in healthcare and in finance.
> >
> > For these types of datasets we provide additional results on both the MIMIC dataset and the income dataset.  We provide these results at the end of the Appendix.  We consider both of these datasets as part of the tabular domain where GAMs can achieve close to state-of-the-art performance.  Therefore, we train a GAM both using the traditional method and using our introduced method.  Generally speaking, the most salient difference is that the InstaSHAP-trained GAMs demonstrate smaller confidence intervals, implying a more consistent interpretation of the data compared to traditional GAM approaches.  This is especially apparent on features which are more correlated (e.g. minimum and maximum blood pressure; capital gains and capital losses).  Although a direct causal relationship still cannot be inferred, the stable insights of our purified GAM models are seen by the authors as a step towards this objective.
> >
> >
> > [1] “From Shapley Values to Generalized Additive Models and back” Sebastian Bordt, Ulrike von Luxburg. 2023.
> >
> > [2] “Feature relevance quantification in explainable AI: A causal problem” Dominik Janzing et al. 2020.

---

> > > ### Comment · Reviewer_pfXM · 2024-11-21
> > >
> > > thanks for clarifying!

---

> > > > ### Author Response · Authors · 2024-11-21
> > > >
> > > > Thank you again for your very insightful review.  It is sincerely hoped that these additional responses and experiments were able to thoroughly address the concerns and questions you brought up.  If after these responses, you still had some lingering concerns or some follow-up questions, we would be happy to try to further address them.  In the context of a majority of the other reviewers having overblown confidence on their understanding of the work, it is hoped we can have a diligent conversation to completely cover the potential concerns about shortcomings of the work.  We apologize to put this burden on you, but simply put it does not seem like other reviewers put as considerable of thought into their responses.  Hence, we again thank you for your considerate review and hope to ask if we have been able to sufficiently address all of the concerns from the original review.  Moreover, if any follow-up concerns exist, we would be happy to attempt to address them.

---

### Meta-Review · Area_Chair_JE8p · 2024-12-20

**Metareview:**

This paper introduces an interpretability method that aims to extend SHAP to account for interaction effects by importing tooling from generalized linear models. The reviewers appreciated the creativity and found the method significant. Concerns relate to the scope of empirical evaluation, the clarity of tradeoffs relative to SHAP, and the relationship to other work in explainability. However, these do not ultimately undermine the conclusion that the paper is worth publishing.

**Additional Comments On Reviewer Discussion:**

See above. As a secondary note, I would encourage the authors to be less belligerent in response to reviewer comments.

---

### Decision · Program_Chairs · 2025-01-22

Accept (Poster)